EMBO
Molecular Medicine

# Fragmentation patterns and personalized sequencing of cell-free DNA in urine and plasma of glioma patients

Florent Mouliere[1,2,3,*,†] (iD), Christopher G Smith[1,2,†] (iD), Katrin Heider[1,2], Jing Su[1,2], Ymke van derPol[3] (iD), Mareike Thompson[1,2], James Morris[1,2], Jonathan C M Wan[1,2], Dineika Chandrananda[1,2,§,¶] (iD), James Hadfield[1,2,††], Marta Grzelak[1,2], Irena Hudecova[1,2], Dominique-Laurent Couturier[1,2], Wendy Cooper[1,2] (iD), Hui Zhao[1,2], Davina Gale[1,2] (iD), Matthew Eldridge[1,2], Colin Watts[4,‡‡], Kevin Brindle[1,2,**,‡] (iD), Nitzan Rosenfeld[1,2,***,‡] (iD) & Richard Mair[1,2,4,****,‡] (iD)

## Abstract

Glioma-derived cell-free DNA (cfDNA) is challenging to detect using liquid biopsy because quantities in body fluids are low. We determined the glioma-derived DNA fraction in cerebrospinal fluid (CSF), plasma, and urine samples from patients using sequencing of personalized capture panels guided by analysis of matched tumor biopsies. By sequencing cfDNA across thousands of mutations, identified individually in each patient's tumor, we detected tumor-derived DNA in the majority of CSF (7/8), plasma (10/12), and urine samples (10/16), with a median tumor fraction of $6.4 \times 10^{-3}$, $3.1 \times 10^{-5}$, and $4.7 \times 10^{-5}$, respectively. We identified a shift in the size distribution of tumor-derived cfDNA fragments in these body fluids. We further analyzed cfDNA fragment sizes using whole-genome sequencing, in urine samples from 35 glioma patients, 27 individuals with non-malignant brain disorders, and 26 healthy individuals. cfDNA in urine of glioma patients was significantly more fragmented compared to urine from patients with non-malignant brain disorders ($P = 1.7 \times 10^{-2}$) and healthy individuals ($P = 5.2 \times 10^{-9}$). Machine learning models integrating fragment length could differentiate urine samples from glioma patients (AUC = 0.80–0.91) suggesting possibilities for truly non-invasive cancer detection.

**Keywords** cell-free DNA; circulating tumor DNA; fragmentomics; gliomas; liquid biopsy

**Subject Categories** Biomarkers; Cancer

## Introduction

Primary brain tumors, which are diagnosed in over 260,000 patients worldwide annually (Wesseling & Capper, 2018), have a poor prognosis and lack effective treatments. Better methods for early detection and identification of tumor recurrence may enable the development of novel treatment strategies. The development of new treatments would also benefit from minimally invasive methods that characterize the evolving glioma genome (Brennan *et al*, 2013; Westphal & Lamszus, 2015). DNA analysis in liquid biopsies has the potential to replace or supplement current imaging-based monitoring techniques, which have limited effectiveness, and to provide the genomic information required for precision medicine while reducing the morbidity associated with repeated biopsy (Mouliere *et al*, 2014; Kros *et al*, 2015; Westphal & Lamszus, 2015). However, cell-free tumor DNA (ctDNA) is extremely challenging to detect in the plasma of patients with brain tumors as its fractional concentration (mutant allele fractions, MAF) is low and appears to be in the same range as that observed in plasma of patients with early-stage

1  Cancer Research UK Cambridge Institute, University of Cambridge, Cambridge, UK
2  Cancer Research UK Major Centre – Cambridge, Cancer Research UK Cambridge Institute, Cambridge, UK
3  Amsterdam UMC, Vrije Universiteit Amsterdam, Department of Pathology, Cancer Centre Amsterdam, Amsterdam, The Netherlands
4  Division of Neurosurgery, Department of Clinical Neurosciences, University of Cambridge, Cambridge, UK
  *Corresponding author. Tel: +31 20 4442405; E-mail: f.mouliere@amsterdamumc.nl
  **Corresponding author. Tel: +44 1223 769647; E-mail: kmb1001@cam.ac.uk
  ***Corresponding author. Tel: +44 1223 769769; E-mail: nitzan.rosenfeld@cruk.cam.ac.uk
  ****Corresponding author. Tel: +44 1223 336946; E-mail: richard.mair@cruk.cam.ac.uk
  †These authors contributed equally to this work as first authors
  ‡These authors contributed equally to this work as senior authors
  §Present address: Peter MacCallum Cancer Centre, Melbourne, Vic., Australia
  ¶Present address: Sir Peter MacCallum, Department of Oncology, The University of Melbourne, Melbourne, Vic., Australia
  ††Present address: Precision Medicine, R&D Oncology Unit, AstraZeneca, Cambridge, UK
  ‡‡Present address: Institute of Cancer Genomics Science, University of Birmingham, Birmingham, UK

carcinomas (Bettegowda *et al*, 2014; Zill *et al*, 2018). Reported detection rates for ctDNA in plasma of glioma patients are typically around 15–30% (Bettegowda *et al*, 2014). Although higher rates of detection have been claimed, the high frequency of alterations resulting from clonal hematopoiesis may confound these results (Zill *et al*, 2018; Pan *et al*, 2019; Piccioni *et al*, 2019). In addition to plasma, ctDNA has been detected in urine for some cancer types, however, this has been limited largely to urothelial cancers, or patients with advanced cancers and high plasma tumor fraction (Husain *et al*, 2017; Patel *et al*, 2017; Bosschieter *et al*, 2018; Dudley *et al*, 2019; Hentschel *et al*, 2020). Cerebrospinal fluid (CSF) has been proposed as an alternative medium for brain tumor ctDNA analysis (De Mattos-Arruda *et al*, 2015; Pan *et al*, 2015, 2019; Wang *et al*, 2015; Pentsova *et al*, 2016; Mouliere *et al*, 2018b; Seoane *et al*, 2019), however, detection sensitivity has remained poor in previous analyses (CSF detected in 42/85 patients, 49.4%) (Miller *et al*, 2019). In addition, CSF sampling via lumbar puncture is an invasive and painful procedure for patients and requires skilled medical staff, which severely limits its use for research, diagnosis, and repeat sampling (Hasbun *et al*, 2001; Engelborghs *et al*, 2017).

Here, using multi-region tumor analysis and tumor-guided deep sequencing in body fluids, we detected and quantified tumor-derived DNA in the majority of urine, plasma, and CSF samples from primary brain tumor patients. This allowed us to measure the concentrations of tumor-derived cfDNA in those body fluids, with a sensitivity not previously attained. Importantly, when using this sensitive technique, we found that detection rate of tumor-derived DNA in urine or plasma was equivalent to that in CSF, despite a higher relative tumor fraction in the latter. Thus, for some applications, the use of invasive CSF sampling could be replaced by sampling of other body fluids. Based on previous data demonstrating the utility of cell-free DNA (cfDNA) fragmentation patterns (Mouliere *et al*, 2018a; Mouliere *et al*, 2018b; van der Pol & Mouliere, 2019), we used a sequencing approach that preserves the structural properties of ctDNA. This allowed us to determine the size profile of mutant ctDNA in matched CSF, plasma, and urine samples from glioma patients. Analyzing urine fragmentation in samples from five patients with low-grade glioma (LGG) and 30 with high-grade glioma (HGG), and 53 individuals without glioma, we demonstrated that urine samples from glioma patients could be identified by analyzing specific fragmentation patterns from shallow whole-genome sequencing (sWGS, < 1× coverage) data using machine learning classifiers.

## Results

We recruited 35 glioma patients (30 HGG, five LGG). Among the five LGG, three were diffuse astrocytoma, one was an oligodendroglioma and one a pilocytic astrocytoma. Among the 30 HGG, 29 were glioblastomas (GBM) and one was an anaplastic oligodendroglioma (AO; Table EV1). In addition, we collected urine samples from 26 healthy volunteers and 27 patients with other pathologies of the brain or central nervous system (CNS; Fig 1 and Table 1). Body fluid samples were analyzed using two sequencing-based approaches: patient-specific hybrid-capture panels and sWGS.

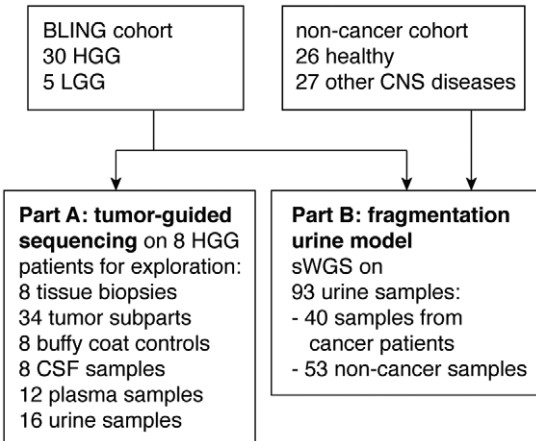

**Figure 1. Study design.**

Schematic of ctDNA detection in matched CSF, plasma and urine samples from HGG and LGG patients using INVAR and/or sWGS. CNS, central nervous system; CSF, cerebrospinal fluid; HGG, high grade glioma; LGG, low grade glioma; sWGS, shallow whole-genome sequencing.

### Sensitive detection and quantification of tumor DNA using patient-specific panels

First, we developed a tumor-guided sequencing assay to determine whether ctDNA can be detected in the bio-fluids of eight patients, including seven with primary GBM and one with AO. From these eight patients, we analyzed 78 samples, including 34 tumor subparts (3–6 per patient), buffy coat, CSF, plasma, and urine (Fig 1 and Table EV1, Table EV2). Initially, DNA from 34 tumor subparts was analyzed individually for each tumor specimen by whole exome sequencing (WES) at an average of 160× coverage, resulting in identification of 5,777 single nucleotide variants (SNV) at an average of 723 SNV per patient (range 266–1,105; Dataset EV1 and Appendix Fig S1). Mutations were initially called by analysis of individual tumor subparts, compared to a non-tumor (buffy coat) sample from the same individual. Next, we merged reads across tumor subparts, thus increasing total sequencing depth, and improving sensitivity for the detection of low abundance mutations. Merging data from the tumor subparts resulted in the detection of 5,517 SNV at an average of 689 SNV per patient (range 215–1,108, Dataset EV2). An average of 21% of the detected SNV were shared between the two mutation calling approaches. 4,015 SNV (54%) were detected only by analysis of the merged data. SNV from both WES approaches were combined for designing the hybrid-capture sequencing panels (7,336 SNV in total after removing overlapping loci). In addition to this tumor-specific list of SNV, the panels were supplemented by comprehensive coverage of the 52 most frequently mutated genes in glioma (Brennan *et al*, 2013).

Sequencing data from the body fluids were analyzed using INVAR (INtegration of VAriants Reads), a recently developed pipeline (Wan *et al*, 2020) that combines locus-based noise filtering, strand selection, and enrichment of mutant fragments using biological characteristics of ctDNA (Appendix Figs S2 and S3). The analytical sensitivity and specificity of INVAR have been defined previously (Wan *et al*, 2020). Using hybrid-capture sequencing panels, we aimed for > 600× coverage for plasma and urine sample

**Table 1. Patient demographics.**

| Parameter | Parameter value |
|---|---|
| Age, years | Mean (range) |
| Overall | 54 (23–87) |
| Other pathologies | 59 (36–87) |
| Healthy | 41 (23–61) |
| Cancer | 61 (24–79) |
| Gender | n |
| Male (total) | 44 |
| Other pathologies | 12 |
| Healthy | 14 |
| Cancer | 18 |
| Female (total) | 44 |
| Other pathologies | 15 |
| Healthy | 12 |
| Cancer | 17 |
| Patient group | n |
| Cancer | 35 |
| Other CNS pathologies | 27 |
| Healthy | 26 |
| Subtype | n |
| HGG | 30 |
| LGG | 5 |
| Aneurysm | 9 |
| Radiculopathy | 7 |
| Myelopathy | 4 |
| Hydrocephalus | 3 |
| Arachnoid cyst | 1 |
| Cavernoma | 1 |
| Hemifacial spasm | 1 |
| Parkinson's disease | 1 |
| IDH (in the cancer cohort) | n |
| Mutant | 5 |
| Wild type | 30 |

Study demographics for gliomas and controls.

sequencing, and $>100\times$ coverage for CSF as its tumor-derived mutant allele fraction has previously been reported to be much higher (Mouliere *et al*, 2018b; Miller *et al*, 2019) (Fig 2A and Dataset EV3). After filtering of the tumor mutations by the INVAR algorithm, sequencing data covering 6,383 unique tumor-derived mutation loci were retained for further analysis (Fig 2B). After applying the capture panel to the body fluids, we observed 366 patient-specific tumor-derived mutations in CSF, 31 in plasma and 67 in urine samples.

Across all fluid samples analyzed (including baseline pre-treatment, post-surgery and follow-up) ctDNA was detected using INVAR in 7/8 CSF samples, 10/12 plasma samples, and 10/16 urine samples (Fig 2C). For three patients, samples of both urine and plasma were obtained 6 months following surgery in addition to the

baseline samples collected immediately prior to surgery (Table EV2). The ctDNA tumor fraction was estimated as an integrated mutant allele fractions (IMAF), which indicated for each sample the average fraction of reads covering target loci that carried a mutant allele that was identified in the same patient's tumor specimens. The mean tumor-derived fraction (IMAF) was $3.1 \times 10^{-5}$ for plasma, $4.72 \times 10^{-5}$ for urine, and $6.4 \times 10^{-3}$ for CSF (Fig 2C). In CSF, more mutations were detected and with a greater number of mutant reads, than in matched plasma and urine samples (Fig 2D). In the fluid samples collected pre-surgery, ctDNA was detected using INVAR in CSF from 7/8 patients, plasma from 7/8 patients, and urine from 6/8 patients (Fig 2E). IMAFs were on average 243-fold lower in plasma samples compared to matched CSF samples (ratios ranged from 9-fold to 1,343-fold) and 389-fold lower in urine samples compared to matched CSF samples (ratios ranged from 50-fold to 834-fold). Tumor-derived DNA was detected 6 months post-surgery in both plasma and urine samples for 2/3 patients (Table EV2). Contrast agent-enhanced $T_1$-weighted MRI demonstrated that the two patients in which ctDNA was detected 6 months after surgery had residual or recurrent disease, whereas the patient with no detected ctDNA had no evidence of recurrence (Fig 2F).

Of the 52 genes frequently mutated in glioma, mutations were detected in 41/52 genes in tumor tissue DNA, 14/52 genes in CSF, 6/52 genes in plasma, and 1/52 genes in urine samples (Fig 3A). In the analysis of body fluid samples in these 52 genes, we detected mutation signals in plasma samples from 4/8 cases, which was equivalent to that detected in CSF (4/8 cases, with signal detected in both fluids in three patients). Conversely, mutant signal in these genes was detected in the urine of only 1/8 cases (who also had signal in plasma). No patient had mutant signal detected in all three fluids. Among the tumor-specific mutations detected in bio-fluids, several mutations in genes frequently altered in gliomas were detected (Fig 3A).

Detection of ctDNA in bio-fluids is thought to be affected by intratumoral genomic heterogeneity (De Mattos-Arruda *et al*, 2015). In our dataset, 859 mutations out of 6,383 that passed INVAR filters (13.5%) were shared across multiple tumor subparts in tumor tissue DNA (Fig 3B and Dataset EV3). In CSF, among the 366 mutations detected pre-surgery, 294 of them (80.3%) originated from mutations shared across multiple tumor subparts (Fig 3C). Conversely, of the 31 mutations detected in plasma samples, only 25.8% originated from mutations shared across multiple tumor subparts (Fig 3D). In urine, this fraction was even lower, and only 8.9% of 67 mutations detected originated from shared tumor tissue mutations (Fig 3E). This different representation in body fluids may reflect the high level of intra-tumor heterogeneity, and the different accessibility to bio-fluid spaces of the heterogeneous populations that make up the tumor mass.

**Tumor-derived cfDNA fragments are shorter than non-mutant cfDNA in the CSF, plasma, and urine samples of glioma patients**

Using paired-end sequencing reads from the hybrid-capture panels, we determined the distribution of read lengths (fragmentation patterns) of mutant and non-mutant cfDNA, i.e. reads carrying mutations previously identified in matched tissue and those not carrying mutations, in the CSF (Fig 4A), plasma (Fig 4B), and urine of the eight glioma patients pre-surgery (Fig 4C). Reads carrying

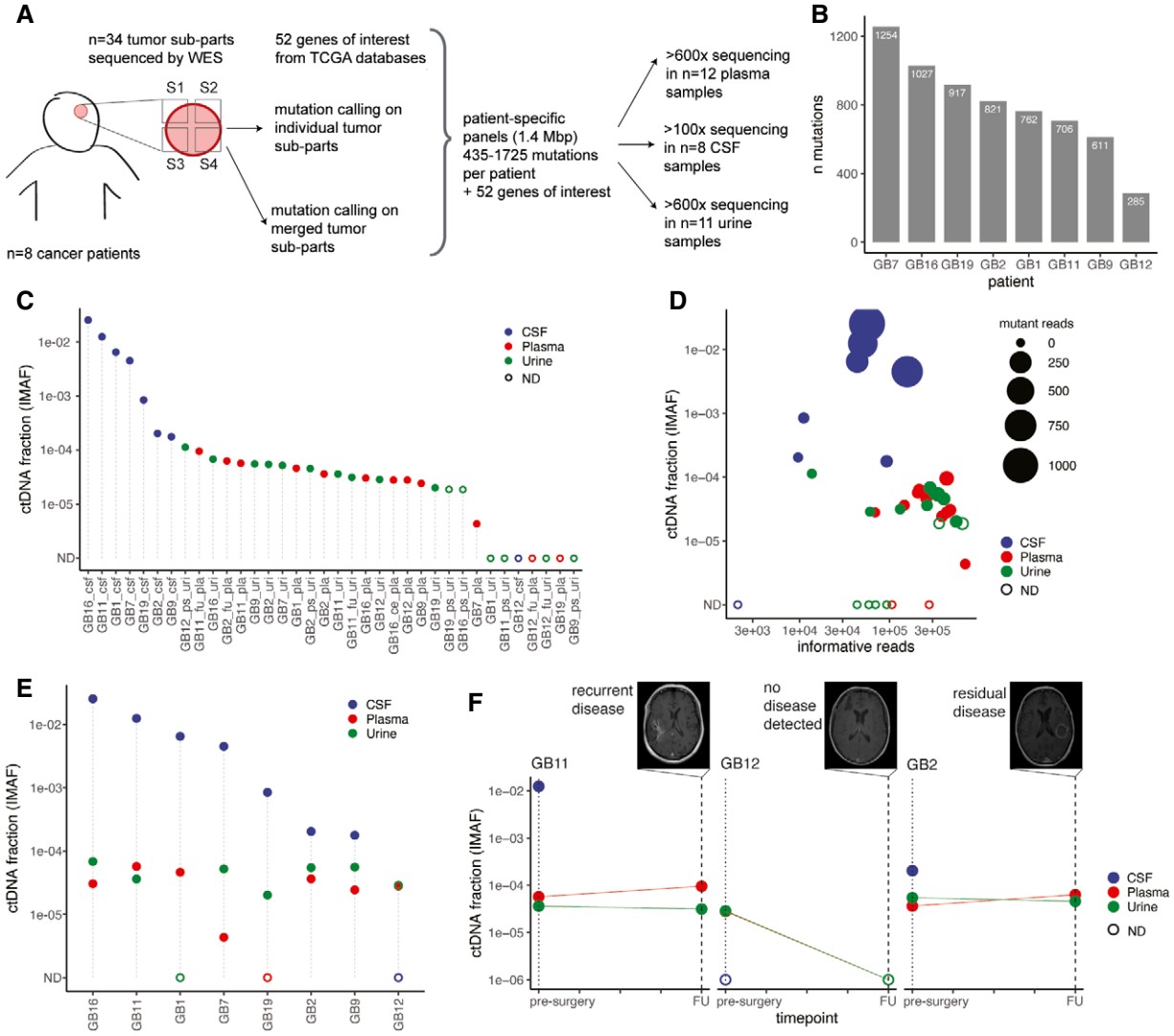

**Figure 2. Detection of ctDNA in CSF, plasma, and urine from glioma patients using patient-specific sequencing panels and INVAR analysis.**

A   Schematic of ctDNA detection in matched CSF, plasma, and urine samples from glioma patients using INVAR (INtegration of VAriants Reads). Depth of sequencing indicated is the mean across the samples analyzed. S1 to S4 indicate tumor subparts.
B   Number of tumor tissue DNA mutations passing INVAR filters for the eight patients included.
C   Estimated ctDNA fractions for the plasma (10/12 detected), CSF (7/8 detected), and urine samples (10/16 detected) collected from 7 patients with primary glioblastoma and one patient (GB12) with anaplastic oligodendroglioma. The ctDNA fraction is expressed as IMAF (Integrated Mutant Allele Fraction). Detected cases are indicated by full circle and non-detected cases as an open circle. ND: non-detected.
D   Estimated tumor DNA DNA fraction (IMAF) in CSF, plasma, and urine depending on the number of mutant reads detected for each samples included and number of informative reads supporting the observation. Detected cases are indicated by full circle and non-detected cases as an open circle. ND: non-detected.
E   Estimated tumor DNA fractions (IMAF) in CSF, plasma, and urine for the matched samples collected at baseline pre-surgery. Detected cases are indicated by full circle and non-detected cases as an open circle. ND: non-detected.
F   IMAF for the CSF, plasma, and urine of the patients with samples collected at 6-month follow-up (n = 3). Matched MRI scans are added for annotation.

tumor-identified mutations represent cfDNA fragments that are highly likely to be derived from the tumor DNA, whereas those without a tumor-identified mutation likely represent a mixture of non-tumor DNA and non-mutated DNA copies from tumor cells. The use of error suppression in the sequencing data analysis results in minimal levels of noise (Wan *et al*, 2020). In the three bio-fluids, we observed a consistent and significant shift toward shorter fragment sizes for mutant cfDNA in comparison with non-mutant

cfDNA: in CSF samples, median size of 148 bp for mutant cfDNA vs 169 bp for non-mutant cfDNA; in plasma samples, 160 vs 169 bp; and in urine samples, 101 vs 133 bp (two-sided Wilcoxon, $P <$ 0.0001 for all three body fluids). Such a shift was described previously for plasma samples of other cancer types (Mouliere *et al*, 2011; Underhill *et al*, 2016; Mouliere *et al*, 2018a; van der Pol & Mouliere, 2019), but has not previously been observed directly in the urine and CSF of patients with gliomas, or other malignancies,

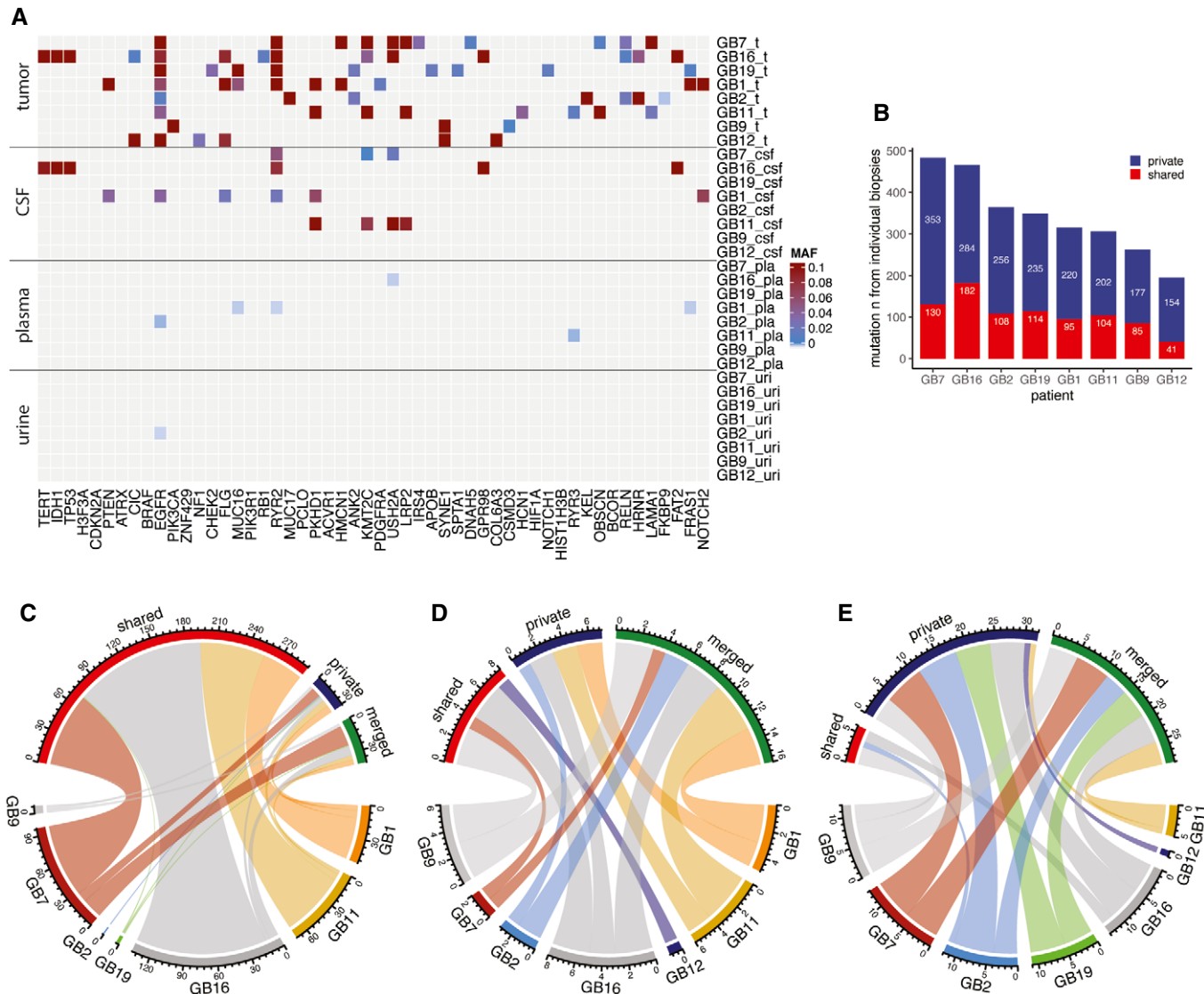

**Figure 3. Detection of ctDNA in plasma, urine, and CSF is affected by intra-tumor heterogeneity in gliomas.**

A    Mutant allele fraction (MAF) measured in tumor, CSF, plasma, and urine samples (on the y axis) for the detected mutations, in the 52 genes that are most frequently mutated in gliomas based on the TCGA databases (on the x-axis), for the eight patients analyzed by targeted sequencing panels. MAF from tumor is derived from exome sequencing. MAF from CSF, plasma, and urine samples are derived from the capture panels sequencing.

B    Number of mutations detected in tumor samples that were observed across several individual tumor biopsies (shared) or private to an individual biopsy.

C–E  Comparison of the number of mutations that were detected in multiple tumor tissue subparts (red), that were private to an individual subpart (blue), or that were rescued by merged calling (green) in CSF (C), plasma (D), and urine (E). Data are shown for samples collected immediately prior to surgery across all eight patients.

by analysis of specifically mutant-derived fragments. We hypothesized that, in a similar way to our previous observations in plasma (Mouliere *et al*, 2018a), the size difference observed in urine could be identified using more scalable methods, to improve ctDNA detection in this non-invasive liquid biopsy without requiring tumor tissue DNA analysis.

**Analysis of cfDNA fragmentation patterns in urine by shallow whole-genome sequencing**

We analyzed the cfDNA fragmentation patterns in 40 urine samples from 35 patients with gliomas (30 HGG and five LGG) collected

pre-treatment with paired-end sWGS (Fig 5A and Appendix Fig S4). We also sequenced urine cfDNA from 53 controls: 26 healthy individuals and 27 patients with other pathologies affecting the central nervous system (cervical myelopathy, cerebral artery aneurysm—both ruptured and unruptured, hydrocephalus and Parkinson's disease; Table 1 and Fig 5A). Baseline urine samples from patients with cancer and other CNS pathologies were collected prior to surgery, and follow-up samples were collected for a subset of the cases (Table EV1). Age and other physiological properties of the cases and controls were collected (Table 1 and Table EV1). All urine samples were collected and processed according to the same protocol and time frame for processing to reduce potential biases due to

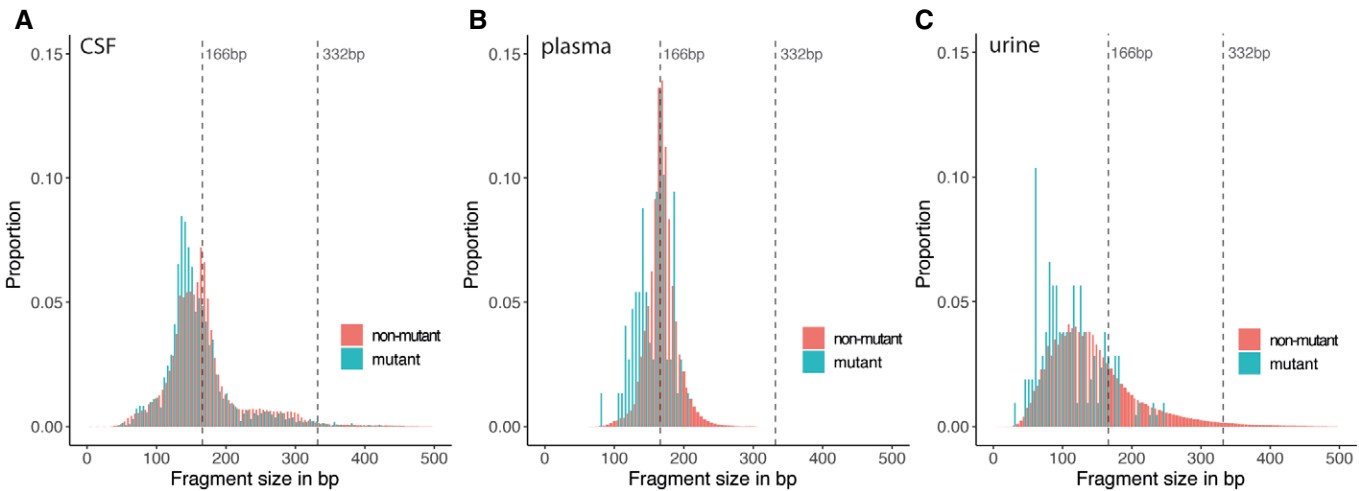

**Figure 4. Mutant cfDNA has shorter fragments than non-mutant cfDNA in the CSF, plasma, and urine samples of glioma patients.**

A–C  Fragment size distributions for mutant (blue) and non-mutant (red) cfDNA reads, determined from the capture sequencing data for CSF samples (A), plasma samples, (B) and urine samples (C).

differences in pre-analytical processing (see Materials and Methods). The mean age of the healthy individuals was lower than for the cancer cases (41 years old and 61 years old, respectively). We therefore evaluated the influence of donor age on the cfDNA fragment size distribution of our cohort of healthy individuals and observed no significant difference (Appendix Fig S5). Of note, the concentration of cfDNA extracted from urines increased from a mean of 4.25 ng/ml in controls to 10.1 ng/ml in glioma patients.

The cfDNA median size distribution in the urine of healthy individuals was 137 bp, 108 bp in the urine of patients with other brain or CNS pathologies, and 101 bp in the urine of glioma patients (Fig 5B). cfDNA in urine of glioma patients was significantly shorter and more fragmented than in urine of healthy individuals (Wilcoxon, $P = 5.2 \times 10^{-9}$) and in urine of patients with other brain pathologies (Wilcoxon, $P = 1.7 \times 10^{-2}$). We calculated the median empirical cumulative distribution function for each type of sample included in the study (Fig 5C). The cumulative distribution indicated that the median fragment size distribution of HGG was significantly different to that of healthy controls (Kolmogorov–Smirnov, distance = 0.476, $P < 0.001$) and of other CNS pathologies (Kolmogorov–Smirnov, distance = 0.287, $P < 0.001$). We analyzed the proportion of fragments in different size ranges and observed that the proportion of fragments between 30 and 60 bp was significantly increased in HGG and LGG cases as compared to healthy controls (Wilcoxon, $P < 0.001$ for HGG and $P < 0.001$ for LGG) and was also increased when compared to patients with other brain or CNS pathologies (Wilcoxon, $P < 0.001$ for HGG and $P = 0.03$ for LGG; Fig 5D).

### Leveraging fragmentation patterns of urine cfDNA for classification of glioma patients from controls

We demonstrated previously that cfDNA fragmentation features could be used to improve the detection of glioma in plasma samples (Mouliere et al, 2018a). Here we explored whether these features in urine could be used to enhance detection of tumor DNA in glioma patients. A predictive analysis was performed using 10 fragmentation features across 93 urine samples (40 samples from 35 cancer cases and 53 samples from 53 non-cancer controls). These 10 fragmentation features were based on the proportion (P) of fragments in the following size ranges in sWGS data from each sample, using 30 bp bins: P(30–60), P(61–90), P(91–120), P(121–150), P(151–180), P(181–210), P(210–240), P(241–270), and P(271–300) (Fig 6A and B). The last feature corresponds to the 10 bp peaks (oscillations) in the distribution of fragment lengths, which have been reported previously (Mouliere et al, 2018a; Mouliere et al, 2018b) and are particularly pronounced in urine samples (Appendix Fig S6A and B). We demonstrated clustering of the data using principal component analysis (PCA; Fig 6C) and t-distributed stochastic neighbor embedding (tSNE; Fig 6D). These indicated that a higher proportion of shorter fragments (< 91 bp) could be indicative of cancer samples (Fig 6C and D). We performed k-means clustering, assuming $k = 2$, and identified a cluster with 29 data points consisting of a high proportion of cancer samples ($n = 27/29$, 94% cancer samples), and a second cluster with 45 data points and a mixture of non-cancer and cancer samples ($n = 13/45$, 28% cancer samples). Analysis of cfDNA fragments using 10 bp bin sizes showed less pronounced clustering (Appendix Fig S7A and B). We tested the individual features and calculated a binary classification to separate "cancer" (HGG and LGG) from "control" samples (healthy and other CNS disease controls; Fig 6E). The feature P30_60 (the proportion of fragments between 30 and 60 bp in length) exhibited the highest classification performance (AUC = 0.885).

Variable selection and the classification of samples as "non-cancer" or "cancer" were performed using logistic regression (LR) and other machine learning models trained and validated on 40 cancer samples and 53 controls (Appendix Fig S8 and Fig 6B). The performance of the models was evaluated using the 10 feature sets, using a double cross-validation scheme and 50 random bootstrap iterations (see Materials and Methods; Fig 6B). Using the SVM

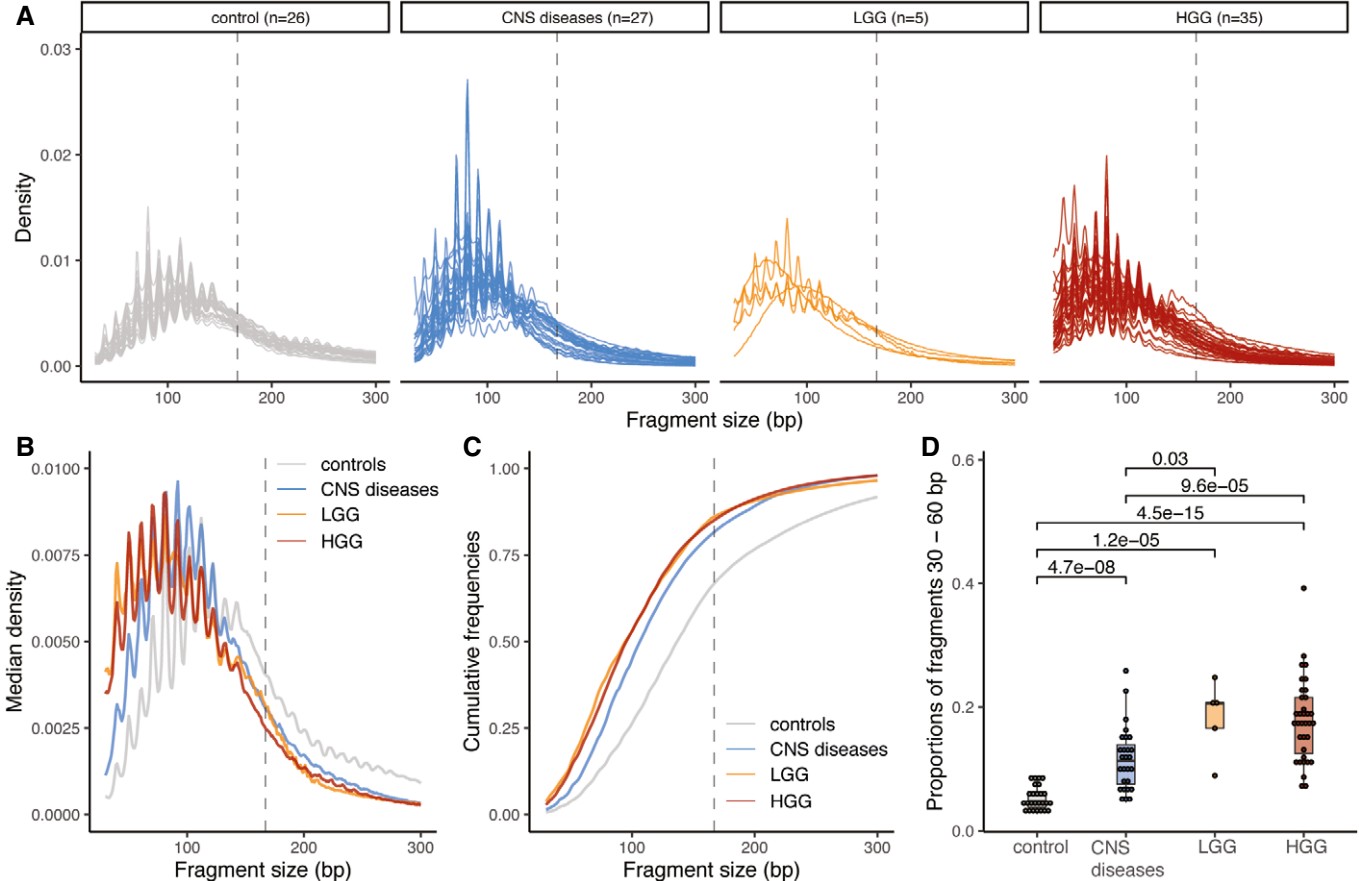

**Figure 5. cfDNA fragmentation patterns are altered in the urine of HGG and LGG patients when compared to healthy controls and other CNS diseases.**

A   Size distribution of urine cfDNA fragments determined from paired-end sWGS (<1× coverage) of 26 healthy controls (in gray), 27 patients with other CNS diseases (cerebral aneurysm, and myeloneuropathy, in blue), five patients with LGG (in orange), and 30 HGG patients (35 samples, in red). Samples from LGG and HGG patients were collected at baseline.

B   Median size distribution of urine cfDNA fragments determined from paired-end sWGS (<1× coverage) for the different patients included in this study (median for each of the groups in part A).

C   Median of the cumulative distribution function of the urine cfDNA fragment sizes of the patients included in this study.

D   Proportion of fragment sizes between 30 and 60 bp in the urine of cfDNA from healthy controls (gray), other non-cancer CNS pathologies (light blue), LGG (orange), and HGG (red). Wilcoxson-test comparing the boxplots are added. Horizontal line within the bars represents median of the underlying population. Boxplot whiskers show 1.5 interquartile range of highest and lowest quartile.

model, we could distinguish non-cancer from cancer samples with a median AUC = 0.80 (range 0.51–1; Fig 6F and G). Sensitivity analyses considering other machine learning methods as classifiers led to similar results in terms of AUC. We compared random forest (RF), support vector machine (SVM), and a binomial generalized linear model with elastic-net regularization (GLMEN) to the LR model. Using the GLMEN model, we could distinguish non-cancer from cancer samples with a median AUC = 0.91 (range 0.76–1; Fig 6F) and a median accuracy = 0.84 (range 0.68–0.95; Fig 6G). The RF model exhibited a median AUC = 0.91 (range 0.76–1) and median accuracy = 0.84 (range 0.68–0.94; Fig 6F and G). The LR model exhibited a median AUC of 0.9 (range 0.70–1) and accuracy = 0.78 (range 0.63–1). Despite the small cohort size (*n* = 93), which might affect the reproducibility of the models with an independent dataset, these results suggest that the cfDNA fragmentation patterns in urine samples may be a useful tool to provide information that can aid in the diagnosis of gliomas.

# Discussion

Tumor-derived DNA has previously been detected in the CSF of patients with glioma and may be helpful for tumor genomic analysis (De Mattos-Arruda *et al*, 2015; Pan *et al*, 2015; Wang *et al*, 2015; Pentsova *et al*, 2016; Mouliere *et al*, 2018b; Miller *et al*, 2019). However, difficulties with longitudinal CSF collection in patients alongside the relative variability in tumor fraction detection may hamper clinical implementation and applicability of CSF analysis. There were different observations reported on the level of detection of ctDNA in plasma of glioma patients (Bettegowda *et al*, 2014; Westphal & Lamszus, 2015; Mouliere *et al*, 2018a; Pan *et al*, 2019). No prior studies had, to our knowledge, explored ctDNA analysis in urine samples from glioma patients.

Here, we have shown that ctDNA can be detected, at very low levels, in the urine and plasma of the majority of patients with high-grade glioma. By tracking a large number of mutations, we

demonstrated that the sensitivity for ctDNA increased and the rate of detection improved, with ctDNA detected in the urine (6/8) and plasma (7/8) of patients with glioma. The median ctDNA fractions in plasma and urine were very low (IMAF of $3.1 \times 10^{-5}$ and $4.72 \times 10^{-5}$, respectively). The size of this cohort ($n = 8$ patients) will need to be increased to demonstrate clinical impact. However, this study reaffirms the high sensitivity and specificity of INVAR demonstrated with other cancer types (melanoma, lung, renal cancers) and early-stage disease (Smith *et al*, 2020; Wan *et al*, 2020). The relative similarity in tumor fraction and detection rate observed between plasma and urine samples was surprising. Prior studies, using animal models, have suggested that the main clearance route of cfDNA from blood is via the liver and not through the kidneys (Gauthier *et al*, 1996; Du Clos *et al*, 1999). The potential alternate mechanisms by which ctDNA could enter the urine will require further investigation. In addition, as a tumor-guided sequencing method, the accuracy of our approach is limited by the nature of the original samples and identified mutations with which we then used to design the capture panel. We have attempted to minimize contamination by mutations originating from clonal expansion in healthy tissues by collecting multiple tumor tissue subparts that were carefully selected during pathological examination. Subsequent tumor-guided sequencing studies will need to select normal tissue DNA and tumor tissue DNA in order to formally exclude the risk of cross-contamination from non-glioma mutations.

While methylation-based detection, cell-free DNA genome-wide fragmentation, tumor-derived mitochondrial DNA, exosomes, vesicles, and tumor-educated platelets have all been proposed as alternative methods for plasma-based detection of glioma-derived mutant DNA, these alternative strategies provide limited information about the tumor genome (Best *et al*, 2015; Moss *et al*, 2018; Mouliere *et al*, 2018a; Shen *et al*, 2018; Mair *et al*, 2019; Nørøxe *et al*, 2019; van der Pol & Mouliere, 2019; Sabedot *et al*, 2021). Through tumor-guided sequencing, we have identified mutations in the plasma and urine of GBM patients and thus demonstrate the potential to track tumor-specific mutations with a high specificity that may be important for monitoring tumor recurrence. In other cancers, ctDNA has been shown to be relatively representative of the clonal architecture of a tumor at a given time (van der Pol & Mouliere, 2019). Here the use of multi-region tumor sampling allowed us to identify shared and private mutations within the primary cancer. We showed that the representation of these mutations varied between the plasma and urine samples, which in our cohort mostly represented private clones, and the CSF, which mostly represented clonal or shared mutations. The higher level of intratumoral heterogeneity in gliomas, and the low levels of release of DNA into plasma, as previously documented (Mouliere *et al*, 2018a; Mouliere *et al*, 2018b; Mair *et al*, 2019), could explain this discordance in representation of spatially distinct clones between the different bio-fluids (van der Pol & Mouliere, 2019). Moreover, a recent study identified two patterns of recurrent disease: a local, predominantly clonal tumor recurrence and a distant predominantly divergent tumor recurrence. It is possible that in the tumors in which we identified mostly private mutations there was significant diffuse disease indicative of divergent tumor evolution already present (Kim *et al*, 2015). Either mechanism would potentially be trackable using our method.

We identified size differences between mutant and non-mutant DNA using tumor-guided sequencing in CSF, plasma, and urine of glioma patients. We analyzed the size distributions of mutant ctDNA by sequencing >435 potentially mutated loci per patient at high depth. This revealed reads that could be unequivocally identified as tumor derived and allowed a direct comparison of fragmentation features of ctDNA as compared to bulk cfDNA. While a powerful technique, a potential limitation of this method is the fact that capture-based sequencing may be biased by probe capture efficiency and therefore may not accurately reflect ratios between tumor and non-tumor DNA, especially for short fragments < 100 bp. Nevertheless, this observation was important as it strongly suggested that ctDNA size shift could be observed in the plasma and the urine of glioma patients. In the case of the former, this agrees with previous data generated using non-capture-based methods.

We complemented this observation by analyzing the genome-wide fragmentation patterns of urine cfDNA in 40 samples from 35 glioma patients using sWGS. We identified cfDNA fragmentation features that could classify urine samples from glioma patients from controls using urine samples, without *a priori* knowledge of somatic aberrations. The median size of cfDNA fragments in urine from control individuals without glioma (137 bp), patients with other CNS diseases (121 bp), and patients with gliomas (101 bp) was different from previous reports on other cancer types (Cheng *et al*,

**Figure 6. cfDNA fragmentation patterns enable classification of glioma patients from controls.**

A  Schematic of the features extracted from the global cfDNA fragmentation patterns of urine samples. 10 features were calculated from the cfDNA fragments size (the proportion of fragments in specific size ranges: P30_60, P61_90, P91_120, P121_150, P151_180, P181_210, P211_240, P241_270, P271_300; and the amplitude of the 10bp oscillations: OSC_10bp).

B  Workflow for the predictive analysis combining the urine fragment size features via LR, RF, SVM, and GLMEN models. sWGS data from 40 urine samples from patients with gliomas and 53 urine samples from controls were split into five subsets for training/validation (80% of the samples) and testing (20% of the samples), according to a 5-fold cross-validation approach and 50 random iterations (see Materials and Methods).

C  Principal component analysis comparing cancer (HGG and LGG) and control samples (healthy and other CNS diseases) using data from the urine fragmentation features. Red arrows indicate features tested during the predictive analysis.

D  tSNE analysis comparing cancer and control samples using data from the same urine fragmentation features.

E  ROC curves for binary classification of cancer and controls for each of the individual fragmentation features analyzed. AUC values are added to the plots.

F  AUC distribution for the unseen test set (samples from patients with gliomas, 40; controls, 53) for four predictive models (LR, GLMEN, RF, and SVM) trained and optimized following the scheme described in (B) and the Materials and Methods section. For each, models are shown the AUC for the 50 iterations. Horizontal line within the bars represent median of the underlying population. Boxplot whiskers show 1.5 interquartile range of highest and lowest quartile.

G  Accuracy were compared for the four classifiers and 50 iterations on the unseen test set of baseline and follow-up samples (19 samples). Horizontal line within the bars represents median of the underlying population. Boxplot whiskers show 1.5 interquartile range of highest and lowest quartile.

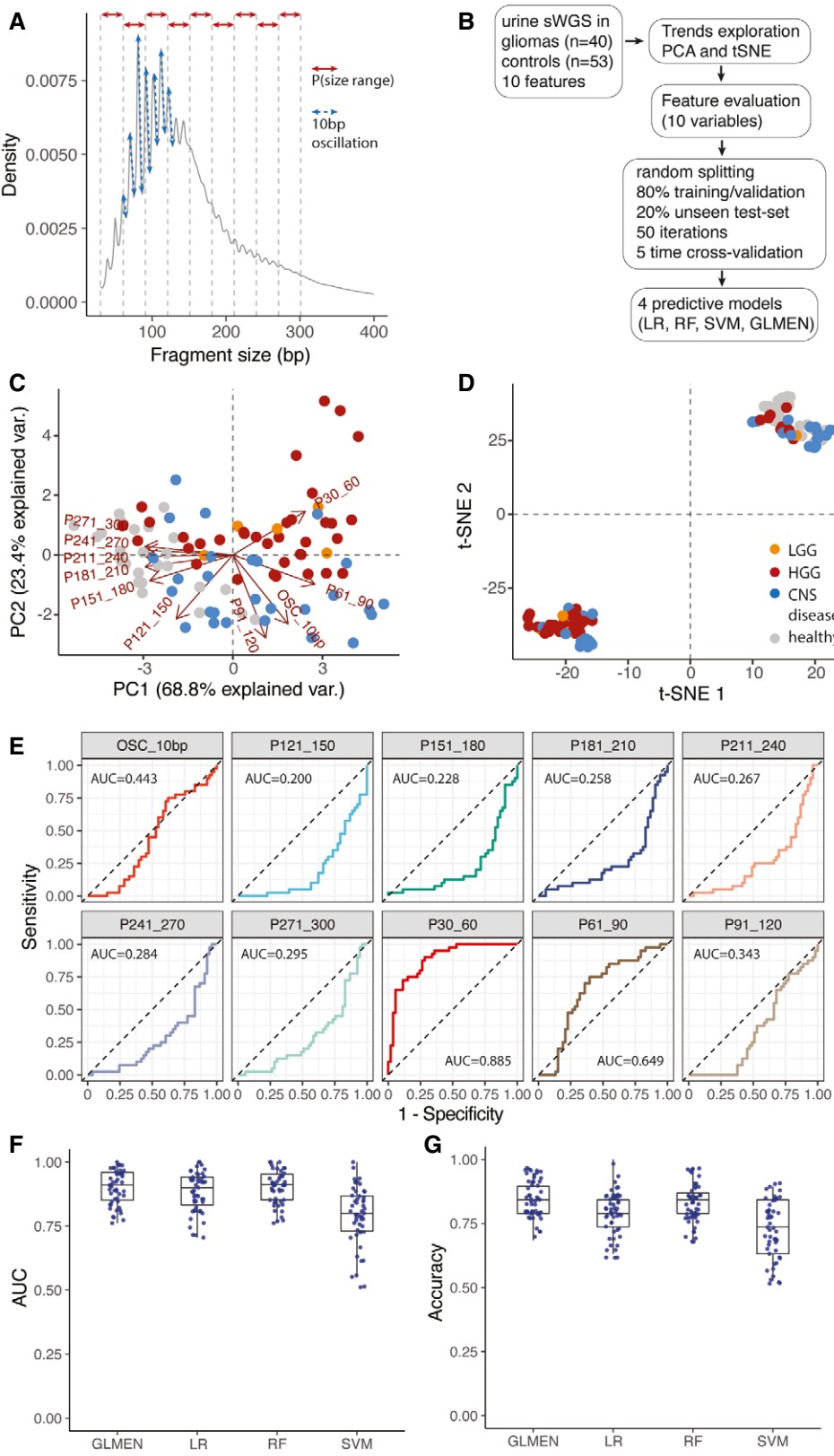

**Figure 6.**

2019; Markus *et al*, 2021). This could indicate that the cfDNA fragmentation profile could be biased depending on the collection procedure and pre-analytical factors. It is also possible that the shortening of cfDNA in the urine of glioma patients compared to controls is due, at least in part, to differences in patient physiology and that this may directly contribute to the detection of a fragmentation-based glioma cfDNA signal in urine. Beyond the tissue of cancer origin, it is likely that urine cfDNA fragmentation might also be influenced by patient physiology (Teo *et al*, 2019), and pre-analytical parameters (Bosschieter *et al*, 2018). We attempted to mitigate for these effects by assessing the effect of age on the cfDNA fragmentation of urine samples, by controlling for the duration of pre-operative fasting, by using standardized sample preparation and DNA isolation and also by assessing the effect of tumor size on detectability. A more in depth analysis of how biological variables impact cfDNA fragmentation in urine samples will be needed in order to conclude the extent to which these factors may lead to different fragmentation patterns in different cohorts.

Such pre-analytical differences notwithstanding, by using a binary classification we observed that the shorter size ranges (P30–60 and P61–90) of cfDNA fragments in urine samples showed larger differences between cancer cases and controls. These size ranges were similar to the size range enriched in mutant cfDNA in urine as observed using tumor-guided capture panels. With four machine learning analyses, we identified and tested 10 size features that can be informative for classifying urine samples as being derived either from healthy individuals or from patients with glioma. The LR, RF, SVM, and GLMEN models correctly classified samples derived from patients with glioma in most of the cases (median AUC = 0.90, median AUC = 0.91, median AUC = 0.80, and median AUC = 0.91, respectively). The GLMEN model correctly identified samples from cancer patients vs samples from controls with a sensitivity of 65% and specificity of 95% in a cohort of 93 urine samples (40 cancer samples and 53 control samples). These results from urine samples from glioma echoed our previous work, which identified 63% of plasma samples from glioma patients with 94% specificity using another RF model based on integration of fragmentation features in plasma cfDNA (Mouliere *et al*, 2018a). Together with other studies that utilize methylation patterns in plasma (Nassiri *et al*, 2020; Sabedot *et al*, 2021), our work suggests that despite a low detection rate of mutations, epigenetic signals (i.e., fragmentation patterns) can be robustly detected in the plasma and also urine of glioma patients.

Our study demonstrates technical feasibility but is limited by the use of double-stranded DNA from urine samples, which is subject to potential biases introduced by the DNA extraction and sequencing methods used (Burnham *et al*, 2016; Bosschieter *et al*, 2018). Analysis of prospective samples that were analyzed blinded to diagnosis would help establish the broader validity and utility of these methods. Nonetheless, we have demonstrated that classification algorithms can utilize information derived from cfDNA fragmentation features to improve the detection of glioma in patients using urine samples. These techniques may therefore provide a method to detect glioma in a truly non-invasive (urine) or minimally invasive (plasma) manner and thus avoiding the morbidity and risk of mortality associated with CSF sampling. Our results encourage further confirmation through the analysis of a larger cohort of both glioma patients and control individuals without cancer.

## Materials and Methods

### Study design

Patients were recruited at Addenbrooke's Hospital, Cambridge, UK as part of the BLiNG (Biopsy of Liquids in New Gliomas) study (REC reference number: 15/EE/0094; Table 1, Tables EV1 and EV2) and the Neurosurgical Research Initiative (18/EE/0172). Patients with suspected GBM on pre-operative contrast-enhanced MRI were chosen, randomly, for participation in the study. The majority of the cohort were sampled during their initial surgery for the new diagnosis of glioma. One patient was sampled at recurrence although this tumor was not found to be hypermutated (GB19) despite treatment with temozolomide in keeping with the relative rarity of hypermutation in IDHwt GBM. Matched tumor tissue, CSF, plasma, urine, and buffy coat samples were collected for eight patients. Urine samples were collected from an additional 27 randomly selected patients. Written informed consent was obtained from the patients; the studies were conducted in accordance with the Declaration of Helsinki and were approved by an Institutional Review Board. Urine from 27 patients with other CNS diseases and 26 healthy individuals were collected using the same collection criteria (Table 1, Tables EV1 and EV2). Informed consent was obtained from all subjects, and all experiments conform to the principles set out in the Declaration of Helsinki and the Department of Health and Human Services Belmont Report.

### Sample collection and preparation

Lumbar puncture was performed immediately prior to craniotomy for tumor debulking. After sterile field preparation, the thecal sac was cannulated between the L3 and L5 intervertebral spaces using a 0.61-mm gauge lumbar puncture needle, and 10 ml of CSF was removed. After collection, CSF, whole blood, and urine samples were immediately placed on ice and then rapidly transferred to a pre-chilled centrifuge for processing. For urine samples, 0.5 M EDTA was added within an hour of collection. Samples were centrifuged at 1,500 *g* at 4°C for 10 min. Supernatant was removed and further centrifuged at 20,000 *g* for 10 min and aliquoted into 2 ml microtubes for storage at −80°C (Sarstedt, Germany). Tumor tissue DNA was extracted and isolated as described previously (Mouliere *et al*, 2018b). Fluids were extracted using the QIAsymphony platform (Qiagen, Germany). Up to 10 ml of plasma, 10 ml of urine and 8 ml of CSF were used per sample. DNA from cancer plasma, urine, and CSF samples was eluted in 90 µl and further concentrated down to 30 µl using a Speed-Vac concentrator (Eppendorf, Germany).

### Sequencing library preparation and WES for tissue DNA

In order to identify patient-specific somatic mutations, we first performed whole exome sequencing (WES) of all tumor tissue and germline buffy coat DNA samples. Fifty nanograms of DNA were fragmented to ˜120 bp by acoustic shearing (Covaris) according to the manufacturer's instructions. Libraries were prepared using the Thruplex DNA-Seq protocol (Rubicon Genomics) with 5× cycles of PCR. Libraries were quantified using quantitative PCR (KAPA library quantification, KAPA biosystems) and pooled for exome capture (TruSeq Exome Enrichment Kit, Illumina). Exome capture

was performed with the addition of i5- and i7-specific blockers (IDT) during the hybridization steps to prevent adaptor "daisy chaining." Pools were concentrated using a SpeedVac vacuum concentrator (Eppendorf, Germany). After capture, 8× cycles of PCR were performed. Enriched libraries were quantified using quantitative PCR (KAPA library quantification, KAPA Biosystems), DNA fragment sizes were assessed by Bioanalyzer (2100 Bioanalyzer, Agilent Genomics), and captures were pooled in equimolar ratio for paired-end next-generation sequencing on a HiSeq4000 (Illumina).

Sequencing reads were de-multiplexed, allowing zero mismatches in barcodes. The reference genome was the GRCh37/b37/hg19 human reference genome—1000 Genomes GRCh37-derived reference genome, which includes chromosomal plus unlocalized and unplaced contigs, the rCRS mitochondrial sequence (AC:NC_012920), Human herpesvirus 4 type 1 (AC:NC_007605) and decoy sequence derived from HuRef, Human Bac and Fosmid clones and NA12878. The sequence data of the patient samples were aligned to the reference genome using BWA-MEM v0.7.15. The duplicate reads were marked using Picard v1.122 (http://broadinstitute.github.io/picard). Somatic SNV and indel mutations were called using GATK Mutect2 (Genome Analysis Toolkit), (https://www.broadinstitute.org/gatk) in tumor-normal pair mode using buffy coat as the normal.

Mutant allele fractions for each single-base locus were calculated with MuTect2 for all bases with PHRED quality ≥ 30. After MuTect2, we applied filtering parameters so that a mutation was called if no mutant reads for an allele were observed in germline DNA at a locus that was covered at least 10×, and if at least four reads supporting the mutant were found in the tumor data with at least one read on each strand (forward and reverse). Variants were annotated using Ensembl Variant Effect Predictor with details about consequence on protein coding, accession numbers for known variants and associated allele frequencies from the 1000 Genomes project.

**Tumor-guided capture sequencing**

Hybrid-based capture for the different body fluids (CSF, plasma, urine) analysis was designed to cover the variants identified above for each patient using the SureDesign software (Agilent). In addition, 52 genes of interest for glioma were included in the tumor-guided sequencing panel based on the TCGA databases. Patients were separated into 2 panels covering all the mutations included for those patients (4 patients per panel). Patients GB1, GB2, GB9, and GB16 were grouped in panel 1, and patients GB7, GB11, GB12, and GB19 were grouped in panel 2. Panel 1 covered in total 526 kbp (5,841 regions) and panel 2 covered 526 kbp (5,701 regions). Panels ranged in size between 1.430 Mb (panel 1) and 1.404 Mb (panel 2) with 120 bp RNA baits. Baits were designed with 5× tiling density, moderately stringent masking and balanced boosting. 99.7% of the targets had baits designed successfully.

Indexed sequencing libraries were prepared using the Thruplex tag-seq kits (Takara). Libraries were captured either in 1-plex for plasma and urine samples or 3-plex for CSF samples (to a total of 1,000 ng capture input) using the Agilent SureSelectXTHS protocol, with the addition of i5 and i7 blocking oligos (IDT), as recommended by the manufacturer for compatibility with ThruPLEX libraries. Custom Agilent SureSelectXTHS baits were used. 13 cycles were used for amplification of the captured libraries. Post-capture libraries were purified with AMPure XT beads, then quantified using

quantitative PCR (KAPA library quantification, KAPA Biosystems), and DNA fragment sizes controlled by Bioanalyzer (2100 Bioanalyzer, Agilent Genomics). Capture libraries were then pooled in equimolar ratios for paired-end next-generation sequencing on a HiSeq4000 (Illumina).

**Capture sequencing analysis and INVAR**

Sequencing reads were de-multiplexed, allowing zero mismatches in barcodes. Cutadapt v1.9.1 was used to remove known 5′ and 3′ adaptor sequences specified in a separate FASTA of adaptor sequences. Trimmed FASTQ files were aligned to the UCSC hg19 genome using BWA-mem v0.7.13 with a seed length of 19. Error suppression was carried out on ThruPLEX Tag-seq library BAM files using CONNOR. The consensus frequency threshold (-f) was set as 0.9 (90%), and the minimum family size threshold (-s) was varied between 2 and 5 for characterization of error rates (Wan et al, 2020). Patient-specific sequencing data consists of informative reads at multiple known patient-specific loci that were identified from tumor sequencing (see above). Because each panel comprised mutations from multiple patients, we could compare mutant allele fractions across loci as a means of error-suppression. Patient's samples could be used as control data for another patient's mutation panels as long as the tumor sequencing did not identify any overlapping mutations. The distribution of signal across loci potentially allows for the identification of noisy loci not consistent with the overall signal distribution. Loci that carried signal in more than 10% of control samples or a mean allele fraction >1% were blacklisted as noisy and removed from the analysis. Each locus was also annotated with trinucleotide error rate, the corresponding tumor allele fraction, fragment size, and whether that locus passed an additional outlier suppression filter as identified by INVAR (INtegration of VAriant Reads), (Wan et al, 2020) (Dataset EV4–EV6). Mutation heat maps were produced in R with the ComplexHeatmap package. Chord diagrams were produced in R with the circlize package.

For each sample, an IMAF (Integrated Mutant Allelic Fraction) was determined across all loci passing pre-INVAR data processing filters with mutant allele fraction at that locus of < 0.25; loci with signal > 0.25 mutant allele fraction were not included in the calculation because (i) loci would not be expected to have such high mutant allele fractions in body fluids of glioma patients (unless they are mis-genotyped SNPs), and (ii) if the true IMAF of a sample is > 0.25, when a large number of loci are tested, they will show a distribution of allele fractions such that detection is still supported by having many low allele fraction loci with signal. Based on the ctDNA level of the sample, the binomial probability of observing each individual locus given the IMAF of that sample was calculated. Loci with a Bonferroni corrected $P$-value < 0.05 (corrected for the number of loci interrogated) were excluded in that sample, thereby suppressing outliers. The detailed calculation of IMAF was previously detailed (Wan et al, 2020).

**sWGS**

Indexed sequencing libraries were prepared using the ThruPLEX-Plasma Seq kit (Rubicon Genomics). Libraries were pooled in equimolar amounts and sequenced to < 0.4× depth of coverage on a HiSeq 4000 (Illumina) generating 150-bp paired-end reads.

Sequence data were analyzed using an in-house pipeline that consists of the following; paired end sequence reads were aligned to the human reference genome (GRCh37) using BWA-mem following the removal of contaminating adapter sequences. PCR and optical duplicates were marked using MarkDuplicates (Picard Tools) feature, and these were excluded from downstream analysis along with reads of low mapping quality and supplementary alignments. When necessary, reads were down-sampled to 10 million in all samples for comparison purposes.

### Fragmentation feature analysis

The preliminary analysis was carried out on 93 samples (40 cancers and 53 non-cancer controls). For each sample, the following features were calculated from sWGS data: P(30–60), P(61–90), P(91–120), P(121–150), P(151–180), P(181–210), P(211–240), P(241–270), and P(271–300). The data were arranged in a matrix where the rows represent each sample and the columns held the aforementioned features with an extra "class" column with the binary labels of "cancer" or "controls." The amplitude of the 10 bp periodic peaks (OSC_10bp) was calculated from the sWGS data as follows: from the samples with clear peaks, the local maxima ("peak") and minima ("valley") in the range 50–140 bp were calculated. The average of their positions across the samples was calculated: (minima: 62, 73, 84, 96, 106, 116, 126, and 137; and maxima: 58, 69, 80, 92, 102, 112, 122, and 134; Appendix Fig S6A and B). To compute the "amplitude statistic," we calculated the sum of the height of the maxima and subtracted the sum of the minima. The larger this difference, the more distinct are the peaks. The height of the $x$ bp peak is defined as the number of fragments with length $x$ divided by the total number of fragments. To define local maxima, we selected the positions $y$ such that $y$ was the largest value in the interval $[y - 2, y + 2]$. The same rationale was used to pick minima. PCA was calculated and visualized in R using the package ggbiplot. The tSNE analysis was performed in R with the Rtsne package using 1,000 iterations, Spearman correlations and a perplexity score of 8. Plots were generated in R using ggplot2. ROC curves were plotted in R with the plotROC package.

### Predictive analysis

The following analysis was carried out in R utilizing RandomForest, and pROC packages and in Python using scikit-learn and H2O Python API modules. The pairwise correlations between the features were calculated to assess multi-collinearity in the dataset (Appendix Fig S8A). Feature importance was analyzed and quantified using a LVQ model. The algorithm was configured to explore all possible subsets of the features. After this pre-processing, all the 10 features were retained for further analysis. The data matrix for the 93 samples (40 cancer samples and 53 controls) were randomly partitioned into five batches of comparable size, four of which were used for training and one was used for testing (80:20 split). For every cross-validation, baseline and follow-up samples of the same patient were randomly distributed in the training set or in the test set. In each of the resulting 5-fold, the training set was split once more using stratified 5-fold cross-validation. This cross-validation scheme was repeated for 10 iterations, yielding 50 iterations in total. Classification of samples as healthy or cancer was performed using logistic regression (LR), random forest (RF), support vector machine

(SVM), and binomial generalized linear models with elastic-net regularization (GLMEN). Predictions on the test set were stored for each of the models 50 folds. To evaluate the performance metric of the models, a ROC curve was calculated for each fold validation and a mean ROC curve were then calculated based on these 50 curves. Mean performance over 50 iterations for precision, recall, accuracy, sensitivity, and specificity were also calculated for each model, and in various scenarios (by selecting all samples, only baseline samples, all features, only 4 features).

### Statistical analysis

All statistics were performed using R (v3.4.3) programming language (www.r-project.org). We also used the ggplot2 (v3.2.0) and ggpubr (v0.2) packages.

## Data availability

Raw sequencing data are deposited at the European Genome-phenome archive (EGAS00001004355; https://ega-archive.org/stud

**The paper explained**

**Problem**

Compared to other disease types, detection of circulating cell-free tumor DNA (cftDNA) in patients with brain tumors, in particular gliomas (GBM), is challenging. While analysis of cftDNA in cerebrospinal fluid (CSF) has improved detection frequencies, this bio-fluid is both difficult to collect and associated with significant discomfort for the patient. As such, it is unlikely that analysis of cftDNA in CSF will be considered as a viable approach for longitudinal sampling going forward. On the other hand, minimally invasive liquid biopsy, in the form of plasma or urine, which do not face these same challenges, could be highly beneficial. However, their use is hampered by the presence of only minute levels of glioma-derived cfDNA signal.

**Result**

First, using tumor-guided sequencing in matched tissue and liquid biopsy, we compared the mutational burden and detection rate of cftDNA in CSF, plasma, and urine from GBM patients. We developed a whole exome sequencing approach that calls mutations that are private to or shared between multiple regions of the same tumor, in doing so affording greater confidence in the mutations calls. These mutations were then used to generate targeted panels for high depth sequencing of CSF, plasma, and urine. By integrating mutation signal across hundreds of mutations, we observed tumor-derived signal in the majority of CSF, plasma, and urine samples.
Then, a second more rapid and cost-effective approach was developed using low coverage WGS. This revealed a possible difference in the fragment sizes of urine cftDNA in cancer patients as compared to healthy individuals. Subsequent application of a machine learning approach to this sequencing data led to the creation of classifiers that demonstrated an 'area under the curve' of between 0.8 and 0.91 for differentiating samples from patients and healthy controls.

**Impact**

The non-invasive nature of plasma and urine may permit more regular and less restrictive monitoring for GBM patients than CSF sampling. While the role of liquid biopsy for diagnosis has been the focus of much attention, both of the methods presented may provide utility in the follow-up setting in combination with imaging.

ies/EGAS00001004355). The INVAR code is available with the following link: http://www.bitbucket.org/nrlab/invar.

**Expanded View** for this article is available online.

## Acknowledgements

We wish to thank for their help and support the Cancer Research UK Cambridge Institute core facilities, in particular bio-repository, bioinformatics, and genomics. cfDNA isolation was performed by the Cancer Molecular Diagnostics Laboratory, which is supported by Cambridge NIHR Biomedical Research Centre, Cambridge Cancer Centre, and the Mark Foundation of Cancer Research. We would also like to acknowledge the support of The University of Cambridge, Cancer Research UK (grant numbers A20240, A29580, A17197, and A16465). The research leading to these results has received funding from the European Research Council under the European Union's Seventh Framework Programme (FP/2007-2013)/ERC Grant Agreement n. 337905. Florent Mouliere is supported by a Dutch Cancer Fund (KWF-12822).

## Author contributions

Concept and design of the study: FM, CGS, NR, and RM; Methodology: FM, CGS, KH, JCMW, and RM; Investigation: FM, CGS, KH, MT, DG, IH, WC, JH, MG, JCMW, IH, WC, and RM; Sample collection: RM and CW; Data analysis: FM and RM; Fragmentation feature design and machine learning analysis: YvdP and FM; Computational analysis: FM, JS, YvdP, HZ, D-LC, CGS, JM, ME, DC, and KH; Writing—original draft: FM and RM; Writing—review and editing: FM, RM, KH, IH, CGS, CW, KB, and NR; Funding acquisition: FM, CW, KB, NR, and RM; Supervision: FM, KB, NR, and RM.

## Conflict of interest

N.R. and D.G. are co-founders, present/former officers or consultants and/or shareholders of Inivata Ltd, a cancer genomics company that commercializes circulating DNA analysis. C.G.S. has consulted for Inivata Ltd. Inivata had no role in the conception, design, data collection, and analysis of the study. Patent applications may be filed describing additional methods described in the manuscript. Other co-authors have no conflict of interests.

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
