## [Review Process File · EMBO Molecular Medicine]

Fragmentation patterns and personalized sequencing of cell-free DNA in urine and plasma of gliomas

Florent Mouliere, Chris Smith, Katrin Heider, Jing Su, Ymke van der Pol, Mareike Thompson, James Morris, Jonathan Wan, Dineika Chandrananda, James Hadfield, Marta Grzelak, Irena Hudecova, Dominique-Laurent Couturier, Wendy Cooper, Hui Zhao, Davina Gale, Matthew Eldridge, Colin Watts, Kevin Brindle, Nitzan Rosenfeld, and Richard Mair

DOI: [10.15252/emmm.202012881](https://doi.org/10.15252/emmm.202012881)

Corresponding authors: Florent Mouliere (f.mouliere@amsterdamumc.nl), Nitzan Rosenfeld (nitzan.rosenfeld@cruk.cam.ac.uk), Richard Mair (richard.mair@cruk.cam.ac.uk)

Review Timeline:

Submission Date:	24th Jun 20
Editorial Decision:	21st Jul 20
Revision Received:	19th Apr 21
Editorial Decision:	7th May 21
Revision Received:	21st May 21
Editorial Decision:	7th Jun 21
Revision Received:	12th Jun 21
Accepted:	14th Jun 21

Editor: Lise Roth

Transaction Report:

21st Jul 2020

Thank you for submitting your work to EMBO Molecular Medicine. We have now heard back from the three referees who agreed to evaluate your manuscript. As you will see below, while the referees mention the interest of the study, they also raise substantial concerns on your work that should be convincingly addressed in a major revision of the present manuscript.

In particular, the tumor-specificity of SNVs detected in body fluids should be established, and the apparently disproportionate impact of the scarce tumor cfDNA on the fragmentation pattern should be carefully addressed.

However, we will not ask for a longitudinal analysis of body fluids as we realize this would require considerable time and effort, and is beyond the scope of the study. This point should nevertheless be discussed in the manuscript.

If you feel you can satisfactorily address these points as well as the other points listed by the referees, you may wish to submit a revised version of your manuscript.

Addressing the reviewers' concerns in full will be necessary for further considering the manuscript in our journal, and acceptance of the manuscript will entail a second round of review. EMBO Molecular Medicine encourages a single round of revision only and therefore, acceptance or rejection of the manuscript will depend on the completeness of your responses included in the next, final version of the manuscript. For this reason, and to save you from any frustrations in the end, I would strongly advise against returning an incomplete revision.

When submitting your revised manuscript, please carefully review the instructions that follow below. Failure to include requested items will delay the evaluation of your revision:

- 1) A .docx formatted version of the manuscript text (including legends for main figures, EV figures and tables). Please make sure that the changes are highlighted to be clearly visible.
- 2) Individual production quality figure files as .eps, .tif, .jpg (one file per figure).
- 3) A .docx formatted letter INCLUDING the reviewers' reports and your detailed point-by-point responses to their comments. As part of the EMBO Press transparent editorial process, the point-by-point response is part of the Review Process File (RPF), which will be published alongside your paper.
- 4) A complete author checklist, which you can download from our author guidelines (<https://www.embopress.org/page/journal/17574684/authorguide#submissionofrevisions>). Please insert information in the checklist that is also reflected in the manuscript. The completed author checklist will also be part of the RPF.

5) Before submitting your revision, primary datasets produced in this study need to be deposited in an appropriate public database (see <https://www.embopress.org/page/journal/17574684/authorguide#dataavailability>). Please remember to provide a reviewer password if the datasets are not yet public. The accession numbers and database should be listed in a formal "Data Availability " section (placed after Materials & Method). Please note that the Data Availability Section is restricted to new primary data that are part of this study.

6) We would also encourage you to include the source data for figure panels that show essential data. Numerical data should be provided as individual .xls or .csv files (including a tab describing the data). For blots or microscopy, uncropped images should be submitted (using a zip archive if multiple images need to be supplied for one panel). Additional information on source data and instruction on how to label the files are available at .

7) Our journal encourages inclusion of *data citations in the reference list* to directly cite datasets that were re-used and obtained from public databases. Data citations in the article text are distinct from normal bibliographical citations and should directly link to the database records from which the data can be accessed. In the main text, data citations are formatted as follows: "Data ref: Smith et al, 2001" or "Data ref: NCBI Sequence Read Archive PRJNA342805, 2017". In the Reference list, data citations must be labeled with "[DATASET]". A data reference must provide the database name, accession number/identifiers and a resolvable link to the landing page from which the data can be accessed at the end of the reference. Further instructions are available at .

8) We replaced Supplementary Information with Expanded View (EV) Figures and Tables that are collapsible/expandable online. A maximum of 5 EV Figures can be typeset. EV Figures should be cited as 'Figure EV1, Figure EV2" etc... in the text and their respective legends should be included in the main text after the legends of regular figures.

- Additional Tables/Datasets should be labeled and referred to as Table EV1, Dataset EV1, etc. Legends have to be provided in a separate tab in case of .xls files. Alternatively, the legend can be supplied as a separate text file (README) and zipped together with the Table/Dataset file. See detailed instructions here: .

9) The paper explained: EMBO Molecular Medicine articles are accompanied by a summary of the articles to emphasize the major findings in the paper and their medical implications for the non-specialist reader. Please provide a draft summary of your article highlighting

10) For more information: There is space at the end of each article to list relevant web links for further consultation by our readers. Could you identify some relevant ones and provide such information as well? Some examples are patient associations, relevant databases, OMIM/proteins/genes links, author's websites, etc...

11) Every published paper now includes a 'Synopsis' to further enhance discoverability. Synopses are displayed on the journal webpage and are freely accessible to all readers. They include a short stand first (maximum of 300 characters, including space) as well as 2-5 one-sentences bullet points that summarizes the paper. Please write the bullet points to summarize the key NEW findings. They should be designed to be complementary to the abstract - i.e. not repeat the same text. We encourage inclusion of key acronyms and quantitative information (maximum of 30 words / bullet point). Please use the passive voice. Please attach these in a separate file or send them by email, we will incorporate them accordingly.

Please also suggest a striking image or visual abstract to illustrate your article. If you do please provide a png file 550 px-wide x 400-px high.

12) As part of the EMBO Publications transparent editorial process initiative (see our Editorial at <http://embomolmed.embopress.org/content/2/9/329>), EMBO Molecular Medicine will publish online a Review Process File (RPF) to accompany accepted manuscripts.

In the event of acceptance, this file will be published in conjunction with your paper and will include the anonymous referee reports, your point-by-point response and all pertinent correspondence relating to the manuscript. Let us know whether you agree with the publication of the RPF and as here, if you want to remove or not any figures from it prior to publication.

I look forward to receiving your revised manuscript.

Yours sincerely,

Lise Roth

Lise Roth, PhD
Editor
EMBO Molecular Medicine

To submit your manuscript, please follow this link:

Link Not Available

Photos 400-800 DPI

*Additional important information regarding figures and illustrations can be found at <http://bit.ly/EMBOPressFigurePreparationGuideline>

***** Reviewer's comments *****

Referee #1 (Remarks for Author):

In the presented manuscript, Smith and colleagues used the cell-free DNA fragmentation sequencing as a biomarker platform. Poor prognosis and lack of effective treatment for patients with glioblastoma underscore an urgent need for better methods for early detection and monitoring of patient response. But such strategies not necessarily provide new insight for the development of novel treatment strategies. So, looking for a cancer cell functional target may decrease a chance to find a good biomarker. On the other hand, potential applications of body fluid other than CSF is significant. Using sequencing approaches that preserve the structural properties of ctDNA, they determined the size profile of mutant ctDNA in matched CSF, plasma, and urine samples from glioma patients. If this is a case, the much broad control cohort for the analysis of plasma and urine will be required to initiate the clinical application of the current study.

Referee #2 (Comments on Novelty/Model System for Author):

Mouliere et al apply deep sequencing of cell-free DNA in plasma, urine, CSF and matched glioma biopsies to evaluate the fragment lengths, abundance and mutational profiles of tumor derived cell-free DNA in plasma and urine. The data is of high quality and exceptional depth, enabling the authors to perform these analyses with greater resolution than previous reports.

I have however reservations about the interpretation of the data.

Referee #2 (Remarks for Author):

Mouliere et al apply deep sequencing of cell-free DNA in plasma, urine, CSF and matched glioma biopsies to evaluate the fragment lengths, abundance and mutational profiles of tumor derived cell-free DNA in plasma and urine. The data is of high quality and exceptional depth, enabling the authors to perform these analyses with greater resolution than previous reports. The scope and design of the study are impressive, and the primary data analysis is sound. I will refrain from providing many technical comments, and instead would like to press the authors on their data interpretation and conclusions.

There are two parts to the analysis: In the first part, the authors assess the fractional abundance of glioma specific cfDNA in blood and plasma, and examine the frequency of detected mutations versus the frequency at which these mutations are detected in the primary tumor. In the second part, the authors analyze fragmentation profiles of tumor specific DNA, and develop machine learning models to detect gliomas from blood and urine via shallow whole genome sequencing. I have two comments related to the interpretation of each part of the analysis.

1. Using sensitive patient-matched exome sequencing, the authors establish a mean tumor-derived fraction of 3.1×10^{-5} in plasma and 4.72×10^{-5} in urine, 243-fold and 389-fold smaller than the ctDNA fraction in the CSF. The authors furthermore find no correlation between the mutant allele fraction in the tumor and the mutant allele fractions in plasma and urine. They do see such correlation in the CSF. Furthermore, the representation of mutations was biased towards shared alleles in the CSF, but tended to be representative of both shared and private mutations within tumor regions. I find these observations very surprising. The authors propose that different regions of the tumor have a different accessibility to bio-fluid spaces to explain these results. Is it possible however that the true fraction of ctDNA from glioma is much lower even than what the authors find, and that the mutations the authors detect in both plasma and the tumor have a non-tumor origin? Clonal expansions in the blood have been shown to contribute significant biological noise in ctDNA analyses. Is it possible that the glioma tumor was contaminated with residual blood, in which clonal hematopoietic expansions are detectable? The same mutations would then also be detectable in plasma. Alternatively, clonal expansions from other solid tissues, outside of the tumor, can contribute DNA to both plasma and the tumor? I think it is difficult to completely rule out a non-tumor origin of mutant alleles detected in both the tumor and plasma/urine, given the very low (1/10,000) proportion of tumor DNA reported here.

2. The authors use shallow whole genome sequencing, and machine learning models trained on details of the distribution of cfDNA fragment lengths to identify glioma from urine. The authors report very high performance of this assay (AUC 0.88-0.91). I find this surprising in light of the very low median tumor fraction observed in this study. How can the tumor meaningfully change the fragmentation profile of the collection of cfDNA in plasma and urine if it makes up such a small fraction? The authors use a pulldown assay to show that mutated cfDNA is indeed shorter, and I have no doubt that this is the case. However, I think it is unlikely that the overall fragment distribution lengths of cell-free DNA (tumor + non-tumor) is meaningfully affected by the presence of a tiny proportion of tumor specific DNA. I appreciate that the authors include samples from non-malignant brain disorders as a control, nonetheless, I don't think they can rule out that the changes in fragment length distributions observed for cancer patients are due to some other physiological effect (lack of mobility, chemotherapy, etc).

Referee #3 (Remarks for Author):

Mouliere and colleagues describe the use of personalized sequencing and cfDNA fragmentation pattern analysis in blood plasma and urine as candidate tools to identify glioma patients and improve follow-up procedures.

The subject is timely and relevant as the study describes a methodology to circumvent the current limitations of glioma liquid biopsy imposed by the paucity of cfDNA in easily accessible body fluids, such as blood or urine.

This methodology exploits a recently developed principle, by which the concomitant study of thousands of tumor-specific single nucleotide variants can compensate for the lack of sequencing depth caused by scarce DNA quantity (Zviran et al. Nat Med, 2020). After WES of tumor tissues, personalized hybrid-capture sequencing panels, targeting specific SNVs and supplemented by the 52 most frequently mutated genes in gliomas, were used to detect the presence of tumor cfDNA in blood and urine. Moreover, this analysis was coupled with assessment of cfDNA fragment sizes with shallow whole genome sequencing, a methodology previously set-up by the same authors and now innovatively applied to urine as well.

The manuscript is written in a technical yet sloppy style, failing to offer conclusions that can be appreciated by a general or clinically-oriented audience (the abstract lacks conclusions at all). Figures, including titles and legends, are carelessly composed, especially in the first part of the manuscript, preventing clear data understanding and interpretation. Overall, the study raises some major issues to be addressed.

A first question concerns the tumor-specificity of the SNVs detected by the capture panel. It is alarming that no driving gene alterations, although often multiple in the same tumor and likely present at high frequency, are ever significantly detected in cfDNA. This requires that validation of at least some of the SNVs found by the panel be attempted by high-sensitivity targeted techniques such as ddPCR.

A second question arises about how the change in cfDNA fragmentation pattern, so remarkable in glioma patients vs. control individuals, can be reconciled with the extreme paucity of tumor DNA in blood and urine. As cf-DNA can derive from many sources, it seems improbable that all cfDNA in glioma patients derives from the tumor, therefore it is unclear how the scanty percentage of tumor cfDNA can cause so evident changes in the cfDNA fragmentation pattern.

Concerning the potential clinical applicability, the personalized sequencing methodology, if proposed as a diagnostic tool, requires an elaborate and expensive analyses on the single patient, hardly feasible in the current, even most advanced clinical contexts. As only information on the tumor burden seems reliably offered (no driver gene alterations are detected by the panel), and as the panel, being customized on the primary tumor, can miss the majority of mutations occurring in the tumor recurring after radio-chemotherapy, it is questionable whether this methodology may provide a net diagnostic improvement compared to standard follow-up with tumor imaging.

The study of DNA fragmentation is potentially more promising for patients' follow-up, but, at this stage, the study fails to provide any example of longitudinal monitoring, such as analysis at pre- and post-surgery, and at different time-points until recurrence. This would be a valuable information, but it would require a considerable additional effort.

Other specific points are as follows:

1. Table S3: the n {degree sign} of listed SNVs is 5777 vs. 8838 reported in the main text. GB2 private mutations: "GB" lacking in rows 604-1445

2. Figure2A:

a. Top histogram: y-axis lacks scale. The histogram displays 16 columns while the sequenced tumor samples are 34. Number of mutations ranges from 435 to 1725 but the height of the last bar is approximately 0. Mutation count is inconsistent with either main text or Table S3.

b. Bottom histogram: y-axis lacks scale. Definitions such as "private clone" (light blue) or "shared

clone" (dark blue), in the figure as well as in the text, seem inappropriate, in the absence of a subclonal composition analysis of the tumor. As an alternative, the terms 'shared mutations', 'partially shared mutations' and 'private mutations' should be used.

3. Figure S1: in panels A and B axes seem wrongly labelled: y-axis should show sequencing depth, while x-axis the fraction of captured target bases {greater than or equal to} depth. Moreover, in tumor samples, 50% of reads correspond to a depth of 100x, while in the main text 160x is indicated.

4. Figure S1D: the collapsed representation of the mutational signature is unconventional, and prevents comparisons with other published signatures. Conventional representation of a 96-bar charts (one for each possible base substitution in their context) should be used.

5. Figure S1F: does the statement "...after individual and merged variant calling..." mean that, single shared mutations are annotated more than once (n individual MAF plus merged MAF)? In this case what is the authors' purpose? In this chart, the nature of variants having a MAF>0.1 is unclear. The use of a logarithmic y-axis scale is recommended.

6. Lines 192-195 refer to samples collected prior and after surgery (6 months), but Figure 2B and C do not display follow-up samples.

7. Lines 207-208. The statement about the finding of several actionable mutations is questionable, as indeed Fig. 3A shows that only a few mutations (most of which not actionable) were detected at very low frequency in urine and plasma. The comment about EGFR therapy failure (211-219) seems out of purpose.

8. Figure 3A "u ri" should be replaced by "uri" or "UR" as in Table S2

9. Figure 3B: surprisingly, a linear regression is shown for data presented in logarithmic scale.

10. Figure 3D: color legend and sample identification is missing. Please clarify, in the legend, that 'variants' correspond to 'mutations' (the term 'variant' is not used or explained in main text).

11. Figure S4 suffers from major problems in data presentation. Color scale does not seem appropriate since (i) the possible highest MAF value is 1 and not 2 as indicated; (ii) only two colors, each representing "high" and "low" frequency variants are used (VAF thresholds should be indicated also in the figure). Moreover, abbreviations used for each sample (suffix 'AF' etc.) should be explained.

12. Figure S4B. For GB7, 6 tumor fractions have been analysed (line 230) but only 3 are represented.

13. Figure 3E and F: authors use different scales (linear and logarithmic) to show CSF and plasma MAF. Although formally correct, this may lead the reader to overestimate the performance of plasma. Therefore, a representation that emphasizes the differences between CSF and plasma should be used.

14. Line 235 refers to both plasma and urine samples but no data on urine can be found either in figure 3 or in figure S4-5.

15. Figure S5. Sample identification (GB7) is missing. The entire figure and its legend are obscure. What does "fraction of patient specific tumor subparts sharing the mutation" mean? In a tumor with 6 samples a chart with 6 "columns" is expected. The data on urine samples mentioned in the main text are not reported. Lines mimicking linear correlations in a chart represented in logarithmic scale should be avoided.

16. Figure S8: the whole picture is unclear. Please try to simplify or use arrows to indicate "peaks" and "valleys".

Referee #1 (Remarks for Author):

In the presented manuscript, Smith and colleagues used the cell-free DNA fragmentation sequencing as a biomarker platform. Poor prognosis and lack of effective treatment for patients with glioblastoma underscore an urgent need for better methods for early detection and monitoring of patient response. But such strategies not necessarily provide new insight for the development of novel treatment strategies. So, looking for a cancer cell functional target may decrease a chance to find a good biomarker. On the other hand, potential applications of body fluid other than CSF is significant. Using sequencing approaches that preserve the structural properties of ctDNA, they determined the size profile of mutant ctDNA in matched CSF, plasma, and urine samples from glioma patients. If this is a case, the much broad control cohort for the analysis of plasma and urine will be required to initiate the clinical application of the current study.

We thank the reviewer for their comments. We have now included 19 additional urine control cases to confirm our results and to demonstrate the potential of our approach. The use of plasma has been demonstrated in our previous work (Mouliere et al., STM, 2018).

Referee #2 (Remarks for Author):

Mouliere et al apply deep sequencing of cell-free DNA in plasma, urine, CSF and matched glioma biopsies to evaluate the fragment lengths, abundance and mutational profiles of tumor derived cell-free DNA in plasma and urine. The data is of high quality and exceptional depth, enabling the authors to perform these analyses with greater resolution than previous reports. The scope and design of the study are impressive, and the primary data analysis is sound. I will refrain from providing many technical comments, and instead would like to press the authors on their data interpretation and conclusions.

There are two parts to the analysis: In the first part, the authors assess the fractional abundance of glioma specific cfDNA in blood and plasma, and examine the frequency of detected mutations versus the frequency at which these mutations are detected in the primary tumor. In the second part, the authors analyze fragmentation profiles of tumor specific DNA, and develop machine learning models to detect gliomas from blood and urine via shallow whole genome sequencing. I have two comments related to the interpretation of each part of the analysis.

We thank the reviewer for their comments.

1. Using sensitive patient-matched exome sequencing, the authors establish a mean tumor-derived fraction of 3.1×10^{-5} in plasma and 4.72×10^{-5} in urine, 243-fold and 389-fold smaller than the ctDNA fraction in the CSF. The authors furthermore find no correlation between the mutant allele fraction in the tumor and the mutant allele fractions in plasma and urine. They do see such correlation in the CSF. Furthermore, the representation of mutations was biased towards shared alleles in the CSF, but tended to be representative of both shared and private mutations within tumor regions. I find these observations very surprising. The authors propose that different regions of the tumor have a different accessibility to bio-fluid spaces to explain these results. Is it possible however that the true fraction of ctDNA from

glioma is much lower even than what the authors find, and that the mutations the authors detect in both plasma and the tumor have a non-tumor origin? Clonal expansions in the blood have been shown to contribute significant biological noise in ctDNA analyses. Is it possible that the glioma tumor was contaminated with residual blood, in which clonal hematopoietic expansions are detectable? The same mutations would then also be detectable in plasma. Alternatively, clonal expansions from other solid tissues, outside of the tumor, can contribute DNA to both plasma and the tumor? I think it is difficult to completely rule out a non-tumor origin of mutant alleles detected in both the tumor and plasma/urine, given the very low (1/10,000) proportion of tumor DNA reported here.

We thank the reviewer for this question on the tumor-derived nature of the variants detected. We can confirm the variants that we detected in CSF, plasma and urine pass our INVAR filters and are therefore tumor-derived. We highlight that the technical demonstration of INVAR, analysis of analytical sensitivity and its potential to capture tumor-specific signal has previously been demonstrated for other cancer types in recent publications from our group (Wan JCM et al, Science Translational Medicine, 2020 and Smith CG et al, Genome Medicine, 2020). Using a spike-in dilution, tumor-derived molecules were detected down to 1 part per million, with various strategies setup to exclude non-tumoral mutations (e.g. noise filtering, fragment size filtration, ... cf Figure S4 – S9 from Wan et al., STM, 2020).

In addition, we observe that mutations detected in the biofluids were also detected in multiple tumor subparts at a high tumor fraction. The same mutations were then detected in CSF, plasma and urine samples further reducing the likelihood of their CHIP origin. The passenger nature of these mutations should not detract from their tumoral origin (as this is the point of our analysis technique).

We are grateful for the opportunity to now develop these points and the potential limitations of the approach in the discussion of the revised manuscript.

2. The authors use shallow whole genome sequencing, and machine learning models trained on details of the distribution of cfDNA fragment lengths to identify glioma from urine. The authors report very high performance of this assay (AUC 0.88-0.91). I find this surprising in light of the very low median tumor fraction observed in this study. How can the tumor meaningfully change the fragmentation profile of the collection of cfDNA in plasma and urine if it makes up such a small fraction? The authors use a pulldown assay to show that mutated cfDNA is indeed shorter, and I have no doubt that this is the case. However, I think it is unlikely that the overall fragment distribution lengths of cell-free DNA (tumor + non-tumor) is meaningfully affected by the presence of a tiny proportion of tumor specific DNA. I appreciate that the authors include samples from non-malignant brain disorders as a control, nonetheless, I don't think they can rule out that the changes in fragment length distributions observed for cancer patients are due to some other physiological effect (lack of mobility, chemotherapy, etc).

Due to the non-targeted nature of sWGS, our approach recovers not only scarce cfDNA fragments with relevant SNVs, but instead the majority of cfDNA fragments released by the tumor and those released by the adjacent brain. Of note, the amount of DNA extracted from urine samples increases from a mean of 4.25 ng/mL in controls to 10.1 ng/mL in glioma patients. Also, none of the patients in this study underwent any treatment e.g chemotherapy or radiotherapy prior to their sampling and so this will not be influencing the cfDNA signal.

Prior work analyzing CpG sites has also demonstrated that in healthy individuals ~5% of cfDNA arises from the CNS and that this may be increased in patients with tumours. Similarly, BBB permeability is affected in patients with gliomas, and thus more brain-related DNA may be released into the circulation. We understand that physiological parameters may also influence cfDNA fragmentation signal and we have attempted to mitigate for physiological effects such as age (Supplementary Figure S7) and fasting status pre-operatively.

Moreover, kidney glomerula are affected by cancer and may filter out different proportions of long and short cfDNA affecting overall fragmentation patterns. Finally, modification of DNA topology or protein complexation could affect the degradation process of cfDNA in the bloodstream/urine altering the overall balance of ct/cfDNA fragments. We have attempted now to account for these points and highlight the potential limitations of this sWGS approach in the discussion of the revised manuscript.

Technologies based on epigenetic alterations have previously demonstrated high sensitivity despite extremely scarce quantities of mutant derived-DNA in circulation (e.g. Nassiri et al, Nature Medicine, 2020; Shen et al, Nature, 2018; Cristiano et al, Nature, 2019; Mouliere et al, STM, 2018). We have also increased our cohort size to include 19 additional non-cancer CNS disease urine samples (e.g. Parkinson's disease). All control cases were age matched with gliomas patients (cf Table 1). The addition of these new controls affirms our observation that cfDNA fragmentation is shorter in the urine of glioma patients. These new samples were also included in the fragmentation pattern comparison (Figure 4) and in our machine learning classification models (Figure 5).

Referee #3 (Remarks for Author):

Mouliere and colleagues describe the use of personalized sequencing and cfDNA fragmentation pattern analysis in blood plasma and urine as candidate tools to identify glioma patients and improve follow-up procedures. The subject is timely and relevant as the study describes a methodology to circumvent the current limitations of glioma liquid biopsy imposed by the paucity of cfDNA in easily accessible body fluids, such as blood or urine.

This methodology exploits a recently developed principle, by which the concomitant study of thousands of tumor-specific single nucleotide variants can compensate for the lack of sequencing depth caused by scarce DNA quantity (Zviran et al. Nat Med, 2020). After WES of tumor tissues, personalized hybrid-capture sequencing panels, targeting specific SNVs and supplemented by the 52 most frequently mutated genes in gliomas, were used to detect the presence of tumor cfDNA in blood and urine. Moreover, this analysis was coupled with assessment of cfDNA fragment sizes with shallow whole genome sequencing, a methodology previously set-up by the same authors and now innovatively applied to urine as well.

We thank the reviewer for their comments.

The manuscript is written in a technical yet sloppy style, failing to offer conclusions that can be appreciated by a general or clinically-oriented audience (the abstract lacks conclusions at all). Figures, including titles and legends, are carelessly composed, especially in the first part

of the manuscript, preventing clear data understanding and interpretation. Overall, the study raises some major issues to be addressed.

We have now re-written important parts of the manuscript and the abstract to take the reviewers comments about clarity into account. All figures, especially in the first part of the manuscript, have been redesigned to increase clarity.

A first question concerns the tumor-specificity of the SNVs detected by the capture panel. It is alarming that no driving gene alterations, although often multiple in the same tumor and likely present at high frequency, are ever significantly detected in cfDNA. This requires that validation of at least some of the SNVs found by the panel be attempted by high-sensitivity targeted techniques such as ddPCR.

Despite a good analytical sensitivity, ddPCR is limited in real plasma samples when the concentration in cfDNA is low (e.g. Bettegowda et al., *STM*, 2013). Moreover, using ddPCR in the glioma context will not confirm the tumor-specificity of the potential SNVs detected, as ddPCR will not enable to differentiate SNVs coming from the tumor (tumor-derived) from the SNVs coming from clonal hematopoiesis (non-tumor derived).

Numerous mutations were detected at high frequency in both the tumor tissue DNA (multiple tumor-subparts) and in the CSF cfDNA (Figure 2). Due to the scarcity of SNVs in plasma and urine, the number of driver variants detected is very low. Capturing driver SNVs is not the point of our approach as we focus on detecting “tumor-derived signal”. The reviewer points out: “This methodology exploits a recently developed principle, by which the concomitant study of thousands of tumor-specific single nucleotide variants can compensate for the lack of sequencing depth caused by scarce DNA quantity (Zviran et al. *Nat Med*, 2020).” The INVAR approach is based on a similar principle and is now validated in a separate document in different pathologies (Wan JCM et al., *Science Translational Medicine*, 2020). In particular figure S4-S9, demonstrates with a spike-in dilution that tumor-derived mutations are accurately detected down to very low tumor fractions (far below the detection limit of single-locus assay like ddPCR). See figure S8 reproduced below. As the technical performance of the method has been previously demonstrated, we are reassured that the mutations passing our analytical filters, and that are detected in the bio-fluids of patients with gliomas, are tumor-specific.

Fig. S8 from Wan et al, STM, 2020: ctDNA dilution series with and without read-collapsing. Spike-in dilution experiment to assess the sensitivity of INVAR. Using error-suppressed data with INVAR, ctDNA was detected in both replicates for all dilutions to 3.6 ppm, and in 2 of 3 replicates at an expected ctDNA allele fraction of 3.6×10^{-7} . Using error-suppressed data of 11 replicates from the same healthy individuals without spiked-in DNA from the cancer patient, no mutant reads were observed in an aggregated 6.3×10^6 informative reads across the patient-specific mutation list.

A second question arises about how the change in cfDNA fragmentation pattern, so remarkable in glioma patients vs. control individuals, can be reconciled with the extreme paucity of tumor DNA in blood and urine. As cfDNA can derive from many sources, it seems improbable that all cfDNA in glioma patients derives from the tumor, therefore it is unclear how the scanty percentage of tumor cfDNA can cause so evident changes in the cfDNA fragmentation pattern.

Due to the non-targeted nature of sWGS, our approach recovers not only scarce cfDNA fragments with relevant SNVs, but also the majority of cfDNA fragments released by the tumor and those released by the adjacent brain. Such a fragmentomic based method, which is analogic by nature, is therefore capable of recovering signal on a much larger base of cfDNA fragments released than a digital method (which in may be more tumor-specific) (Im et al, Trends in Cancer, 2020). Of note, the amount of DNA extracted from urine samples increases from a mean of 4.25 ng/mL in controls to 10.1 ng/mL in glioma patients in our study.

Here we show that our sWGS approach can differentiate tumor signal from non-tumor signal, and that such a method could be used to classify cancer versus controls using urine samples (we previously demonstrated that such an approach could work in plasma samples from glioma patients) (Mouliere et al, STM, 2018).

Technologies based on epigenetic alterations have previously demonstrated good sensitivity despite extremely scarce quantities of mutant derived-DNA in circulation (e.g. Nassiri et al, Nature Medicine, 2020; Shen et al, Nature, 2018; Cristiano et al, Nature, 2019; Mouliere et al, STM, 2018). We have also increased our cohort size to include 19 additional non-cancer CNS disease urine samples (e.g. Parkinson's disease). All control cases were age matched with gliomas patients (cf Table 1). The addition of these new controls affirms our observation that cfDNA fragmentation is shorter in the urine of glioma patients. These new samples were also included in the fragmentation pattern comparison (Figure 4) and in our machine learning classification models (Figure 5).

Prior work using CpG sites has also demonstrated that in healthy individuals ~5% of cfDNA arises from the CNS and this may be increased in patient with tumours. Similarly, BBB permeability is affected in patients with gliomas, and thus more brain-related DNA may be released into the circulation. We understand that physiological parameters may also influence cfDNA fragmentation signal and we have attempted to mitigate for physiological effects such as age (Supplementary Figure S7) and fasting status pre-operatively.

Moreover, kidney glomerula are affected by cancer and may filter out different proportions of long and short cfDNA affecting overall fragmentation patterns. Finally, modification of DNA topology or protein complexation could affect the degradation process of cfDNA in the bloodstream/urine altering the overall balance of ct/cfDNA fragments. We have attempted now to address some of these points in the discussion of the revised manuscript.

Concerning the potential clinical applicability, the personalized sequencing methodology, if proposed as a diagnostic tool, requires an elaborate and expensive analyses on the single patient, hardly feasible in the current, even most advanced clinical contexts. As only information on the tumor burden seems reliably offered (no driver gene alterations are detected by the panel), and as the panel, being customized on the primary tumor, can miss the majority of mutations occurring in the tumor recurring after radio-chemotherapy, it is questionable whether this methodology may provide a net diagnostic improvement compared to standard follow-up with tumor imaging. The study of DNA fragmentation is potentially more promising for patients' follow-up, but, at this stage, the study fails to provide any example of longitudinal monitoring, such as analysis at pre- and post-surgery, and at different time-points until recurrence. This would be a valuable information, but it would require a considerable additional effort.

We have explained in greater details the limitations and potential applications of the two approaches in the discussion. We are not proposing this technique as being suitable for diagnosis and mention this specifically in the discussion. Clinically, this is also less useful due to surgery being the primary treatment for these patients using current management paradigms.

A personalized sequencing strategy might, however, be very well adapted to monitor tumor recurrence in an MRD setting (and multiple references in the literature are confirming the validity of such approach in other cancer type, such as the work by Zviran et al. Nat Med, 2020 previously cited by the reviewer). Studies ongoing in several centres, including our own,

investigating whole genome sequencing utility at diagnosis in several cancers and this approach would also be suitable for glioma. We also know that a significant proportion of patients present with local recurrence and genomic evolutionary trajectories similar to those of the initial tumor (Kim et al 2015, Cancer cell) supporting clinical potential.

Costs for such technology remain high, but are decreasing due to NGS library generation and sequencing development and thus prospective sequencing for biomarkers may be a valid option in the near future. This could either be independently, or in conjunction imaging biomarkers. We also note that numerous liquid biopsy companies are already including personalized detection of ctDNA (with sequencing panel or ddPCR) in their portfolio (e.g. Inivata or Natera).

In contrast, leveraging an epigenetic biomarker, such as methylation or cfDNA fragmentation, may be appropriate for screening, risk stratification and cancer classification (as recently demonstrated by Cristiano et al, Nature, 2019 or Nassiri et al, Nat Med, 2020).

We appreciate the reviewer's perspective on longitudinal results and that our approach would benefit from analysis across a larger cohort of cases. We have performed a power calculation has been performed based on the AUC from our machine learning model to estimate the recruitment time and number of samples to collect to investigate recurrence using urine samples. Such an approach would take ~2 years and is therefore unfortunately beyond the scope of this current study (for which we have aimed to demonstrate feasibility for urine liquid biopsy in glioma).

Other specific points are as follows:

1. Table S3: the n{degree sign} of listed SNVs is 5777 vs. 8838 reported in the main text. GB2 private mutations: "GB" lacking in rows 604-1445
Table S3 is now **Data Source 1** in the revised version of the manuscript. 5777 mutations were called in tissue DNA during the first round of mutation calling (on the individual tumor-subparts), and additional 3061 unique new mutations were detected after merging the individual tumor subparts to increase the sensitivity of the calling to capture clonal variants. After filtering of the mutations by our INVAR algorithm, 6384 unique tumor-derived mutations were retained for further analysis (the table with the list of unique tumor-derived SNVs is now added as **Data source 3**). "GB" has been added to the missing rows.
2. Figure2A: a. Top histogram: y-axis lacks scale. The histogram displays 16 columns while the sequenced tumor samples are 34. Number of mutations ranges from 435 to 1725 but the height of the last bar is approximately 0. Mutation count is inconsistent with either main text or Table S3. b. Bottom histogram: y-axis lacks scale. Definitions such as "private clone" (light blue) or "shared clone" (dark blue), in the figure as well as in the text, seem inappropriate, in the absence of a subclonal composition analysis of the tumor. As an alternative, the terms 'shared mutations', 'partially shared mutations' and 'private mutations' should be used.

Figure 2A is a schematic illustrating and explaining the overall personalized sequencing approach (as explained in the legend). Following the reviewer's recommendation we have decided to clarify this figure subpart as below:

- Figure S1: in panels A and B axes seem wrongly labelled: y-axis should show sequencing depth, while x- axis the fraction of captured target bases {greater than or equal to} depth. Moreover, in tumor samples, 50% of reads correspond to a depth of 100x, while in the main text 160x is indicated.

Figure S1 has been fully redesigned as requested (see below). We confirm that panel A and B were correctly labelled.

Appendix Figure S1: quality control assessment of the WES data from the tumor tissue DNA.

A: Sequencing depth from white blood cell DNA extracted from the buffy coat layer. B: Sequencing depth from the tumor tissue DNA. C: Mutation counts depending on the AF, colored by mutation class. D: Mutation context of the tumor tissue DNA data. Are displayed the unique patient-specific polished filtered data retained by the INVAR algorithm. E: Mutation counts by trinucleotide context, colored by mutation class.

- Figure S1D: the collapsed representation of the mutational signature is unconventional, and prevents comparisons with other published signatures. Conventional representation of a 96-bar charts (one for each possible base substitution in their context) should be used.

A more “conventional” 96-bar chart of the mutational context is now added in the **Figure S1D**. As the previously published works from our group based on the INVAR method are using this data visualization to represent the mutation count by trinucleotide context (Wan et al, STM, 2020 and Smith et al, Genome Medicine, 2020), we have also maintained this visualization in **Figure S1E** from the revised document.

- Figure S1F: does the statement "...after individual and merged variant calling..." mean that, single shared mutations are annotated more than once (n individual MAF plus merged MAF)? In this case what is the authors' purpose? In this chart, the nature of variants having a MAF>0.1 is unclear. The use of a logarithmic y-axis scale is recommended.

We have explained further what we meant with the individual and merged calling in the document. Shared mutations are accounting for duplicates and are annotated once. In particular our method to call and integrate individual and merged tumor subparts is now described in the results section of the document. This approach has been selected for designing the capture panel in order to maximise the number of mutations captured by the panel.

The figure S1F has been now removed in a new version of Figure S1.

- Lines 192-195 refer to samples collected prior and after surgery (6 months), but Figure 2B and C do not display follow-up samples.

Samples were collected 6 months after surgery for 3 patients only (i.e. when available), and immediately post-surgery for another subset of urine cases. These samples have been added to Figure 2C, Figure 2D and detailed further in Figure 2E for the follow-up samples collected 6 months post-surgery (see below).

- Lines 207-208. The statement about the finding of several actionable mutations is questionable, as indeed Fig. 3A shows that only a few mutations (most of which not actionable) were detected at very low frequency in urine and plasma. The comment about EGFR therapy failure (211-219) seems out of purpose.

We agree to mitigate our sentence about “clinically actionable mutations”, as we mostly detect mutations in genes frequently altered in gliomas (based on TCGA data). We have rewritten this sentence as follow: “Amongst the tumor-specific mutations detected in bio-fluids, several mutations in genes frequently altered in gliomas were detected (**Figure 3A**).”

We also agree that our comment about EGFR therapy failure was out purpose and this sentence has been removed.

8. Figure 3A "u ri" should be replaced by "uri" or "UR" as in Table S2
We are now using “uri” in **Figure 3A**.
9. Figure 3B: surprisingly, a linear regression is shown for data presented in logarithmic scale.
This figure subpart has been removed in the new version of the manuscript.
10. Figure 3D: color legend and sample identification is missing. Please clarify, in the legend, that 'variants' correspond to 'mutations' (the term 'variant' is not used or explained in main text).
This figure subpart has been removed in the new version of the manuscript.
11. Figure S4 suffers from major problems in data presentation. Color scale does not seem appropriate since (i) the possible highest MAF value is 1 and not 2 as indicated; (ii) only two colors, each representing "high" and "low" frequency variants are used (VAF thresholds should be indicated also in the figure). Moreover, abbreviations used for each sample (suffix 'AF' etc.) should be explained.
We agree that this supplementary figure was very unclear and of limited added value for the manuscript, and therefore it has been removed.
12. Figure S4B. For GB7, 6 tumor fractions have been analysed (line 230) but only 3 are represented.
This figure has been removed from the revised manuscript.
13. Figure 3E and F: authors use different scales (linear and logarithmic) to show CSF and plasma MAF. Although formally correct, this may lead the reader to overestimate the performance of plasma. Therefore, a representation that emphasizes the differences between CSF and plasma should be used.
Figure 3E and Figure 3F have been modified in the updated version of our manuscript.
14. Line 235 refers to both plasma and urine samples but no data on urine can be found either in figure 3 or in figure S4-5.
This data for urine samples have been added in **Figure 3**. In our revision, **Figure S4 and Figure S5** have been removed from the updated appendix but data can be found in their respective Data Source documents.
15. Figure S5. Sample identification (GB7) is missing. The entire figure and its legend are obscure. What does "fraction of patient specific tumor subparts sharing the mutation" mean? In a tumor with 6 samples a chart with 6 "columns" is expected. The data on

urine samples mentioned in the main text are not reported. Lines mimicking linear correlations in a chart represented in logarithmic scale should be avoided.

We agree that this supplementary figure was unclear and of limited added value for the manuscript, and therefore it has been removed.

16. Figure S8: the whole picture is unclear. Please try to simplify or use arrows to indicate "peaks" and "valleys".

Figure S7A (in the revised manuscript) represents the experimentally determined "peaks" whereas **Figure S7B** represents the "valleys" calculated as described in the Methods section. The optimal "peaks" and "valleys" in specific size range are therefore calculated for each sample by a dot represented in the figure (see Methods). We improved the description of the figure in the figure legends. We also previously defined this approach and method (e.g. Mouliere et al. EMBO Molecular Medicine, 2018).

7th May 2021

Thank you for the submission of your revised manuscript to EMBO Molecular Medicine. We have now received feedback from the two referees who re-reviewed your manuscript.

As you will see from the reports below, both referees acknowledge your efforts to address their initial concerns and recognize that the manuscript has significantly improved. However, referee #2 mentions issues that remain unanswered.

Therefore, we would like you to address the comments raised by this referee (experimentally or by discussing these points), as well as the minor points raised by referee #3. Please be aware that this will be the last chance for you to address these points.

Additionally, please also address the following editorial issues:

- We can accommodate up to 5 keywords, please adjust accordingly.
- All corresponding authors are required to supply an ORCID ID for their name upon submission of a revised manuscript. It is currently missing for Richard Mair. We note that you have together with you, a total of 4 co-corresponding authors. Is that correct? Do you confirm equal contribution of these 4 people, able to take full responsibility for the paper and its content? While there is no limit per se to the number of co-corresponding authors, 3 is rare, 4 even more so, and may not reflect as intended to the community.
- 'Data and code availability' should be renamed 'Data availability'. Please provide a direct link for the EGA data and make sure that the data are public before acceptance of the manuscript.
- Please provide 'The paper explained' section: EMBO Molecular Medicine articles are accompanied by a summary of the articles to emphasize the major findings in the paper and their medical implications for the non-specialist reader. Provide a draft summary of your article highlighting:
 - the medical issue you are addressing,
 - the results obtained and
 - their clinical impact.
- References should list 10 authors before et al. (currently 20 authors et al)
- The Appendix file should contain a table of content.
- Please remove "Data not shown". As per our guidelines, all data referred to in the paper should be displayed in the main or Expanded View figures.
- Every published paper now includes a 'Synopsis' to further enhance discoverability. It includes a short stand first (maximum of 300 characters, including space) as well as 2-5 one-sentences bullet points that summarizes the paper. Please write the bullet points to summarize the key new findings. They should be designed to be complementary to the abstract - i.e. not repeat the same text. We encourage inclusion of key acronyms and quantitative information (maximum of 30 words / bullet point). Please use the passive voice. Please attach these in a separate file or send them by email, we will incorporate them accordingly. Please also suggest a striking image or visual abstract to illustrate your article as a png file 550 px-wide x 400-px high.
- As part of the EMBO Publications transparent editorial process initiative (see our Editorial at <http://embomolmed.embopress.org/content/2/9/329>), EMBO Molecular Medicine will publish online a Review Process File (RPF) to accompany accepted manuscripts. In the event of acceptance, this file will be published in conjunction with your paper and will include the anonymous referee reports, your point-by-point response and all pertinent correspondence relating to the manuscript.

Let us know whether you agree with the publication of the RPF and as here, if you want to remove or not any figures from it prior to publication.

I look forward to receiving your revised manuscript.

With my best wishes,

Lise

Lise Roth, PhD
Editor
EMBO Molecular Medicine

To submit your manuscript, please follow this link:

Link Not Available

Photos 400-800 DPI

*Additional important information regarding figures and illustrations can be found at <https://bit.ly/EMBOPressFigurePreparationGuideline>

***** Reviewer's comments *****

Referee #2 (Remarks for Author):

Mouliere et al have revised their manuscript and responded to the comments from all reviewers. As I wrote in my initial review, the question addressed in this paper is of significant interest, and the authors present a valuable dataset and have performed an impressive series of experiments in which they implemented state-of-the-art measurement technologies. The new version of the manuscript further improves the presentation of the data and results. Nonetheless, I remain unconvinced with the interpretation of two key results in this paper. I cannot recommend this paper for publication in its current form, unless the authors address the below two interpretation issues in

the discussion section of the paper, or with additional experiments that rule out alternative interpretations. I have raised these same issues in my initial review, as did reviewer #3.

First: I still think it is impossible to rule out a non-glioma origin of mutant alleles detected in both the tumor and plasma/urine, given 1) the very low (1/10,000) proportion of tumor cell-free DNA reported here, 2) the lack of correlation between mutant alleles detected in the tumor and mutant alleles detected in plasma/urine, 3) the low frequency of detection of mutations frequently observed in glioma, and 4) the infrequent detection of mutations shared across different parts of the tumor. The authors insist that this observation is due to the "different accessibility to bio-fluid spaces of the heterogeneous populations that make up the tumor mass". I remain unconvinced of this argument. An alternative explanation is that these mutations are not glioma derived but rather due to clonal expansions in other tissues. The authors did not perform exome sequencing on the healthy subjects to rule out that a similar level of mutations is observed in this control group of patients. (it would be good to know the age of the subjects for which the mutations were detected, I was not able to determine this based on the supplemental data tables provided)

Second: the authors use shallow whole genome sequencing and analyses of overall DNA fragment length profiles to identify glioma based cfDNA. I had previously raised the issue that it is unlikely that the overall fragment length distribution is meaningfully affected by the presence of a tiny proportion of glioma specific DNA. Reviewer #3 raised a similar point "A second question arises about how the change in cfDNA fragmentation pattern, so remarkable in glioma patients vs. control individuals, can be reconciled with the extreme paucity of tumor DNA in blood and urine". The authors counter "Due to the non-targeted nature of sWGS, our approach recovers not only scarce cfDNA fragments with relevant SNVs, but also the majority of cfDNA fragments released by the tumor and those released by the adjacent brain". I remain unconvinced of this argument. The non-targeted assay recovers more tumor DNA of course, but it also recovers more DNA from healthy tissues. An alternative explanation here is that the shortening of cfDNA in the plasma of glioma patients compared to controls is due to some other change in patient physiology, it is unlikely the authors are detecting glioma specific DNA in this way.

Referee #3 (Comments on Novelty/Model System for Author):

Concerning the medical impact, the authors provide only proof of concept that an innovative but complex and expensive methodology to detect glioma DNA in plasma and urine is feasible.

Referee #3 (Remarks for Author):

The authors adequately addressed the points raised by this reviewer and rewrote the manuscript in order to improve clarity and efficacy.

Please amend this inconsistency:

-line 838: AUC = 0.921 while in Fig. 6E AUC = 0.885.

Please check nomenclature correspondence between column A in Table EV1 and in Table EV2 (Bling vs. GB)

Referee #2 (Remarks for Author):

Mouliere et al have revised their manuscript and responded to the comments from all reviewers. As I wrote in my initial review, the question addressed in this paper is of significant interest, and the authors present a valuable dataset and have performed an impressive series of experiments in which they implemented state-of-the-art measurement technologies. The new version of the manuscript further improves the presentation of the data and results. Nonetheless, I remain unconvinced with the interpretation of two key results in this paper. I cannot recommend this paper for publication in its current form, unless the authors address the below two interpretation issues in the discussion section of the paper, or with additional experiments that rule out alternative interpretations. I have raised these same issues in my initial review, as did reviewer #3.

First: I still think it is impossible to rule out a non-glioma origin of mutant alleles detected in both the tumor and plasma/urine, given 1) the very low (1/10,000) proportion of tumor cell-free DNA reported here, 2) the lack of correlation between mutant alleles detected in the tumor and mutant alleles detected in plasma/urine, 3) the low frequency of detection of mutations frequently observed in glioma, and 4) the infrequent detection of mutations shared across different parts of the tumor. The authors insist that this observation is due to the "different accessibility to bio-fluid spaces of the heterogeneous populations that make up the tumor mass". I remain unconvinced of this argument. An alternative explanation is that these mutations are not glioma derived but rather due to clonal expansions in other tissues. The authors did not perform exome sequencing on the healthy subjects to rule out that a similar level of mutations is observed in this control group of patients. (it would be good to know the age of the subjects for which the mutations were detected, I was not able to determine this based on the supplemental data tables provided).

We thank the Reviewer for this comment. We agree that despite our best efforts we cannot rule out that we solely capture DNA of glioma origin in the tumor, CSF, plasma and urine samples. In our opinion performing WES on healthy sample will only give us an approximative information regarding the error-rate in these samples, and will not rule out completely potential contamination by mutations from clonal expansion in other tissues. Sequencing biopsies of normal tissues from the same patients is not possible anymore and will require recruitment of new patients, which is beyond the scope of this study. Such an approach would be as one could not perform representative sampling of every tissue across a patient.

As the Reviewer points out above, the proportion of ctDNA is very low in these samples. Therefore, the specific alleles that are detected in any sample are expected to be a random small subset of the spectrum of mutations present in the tumor – the premise of the INVAR approach is that we will detect a small number of those mutations by sampling a large number of tumor-mutated loci. Therefore, low levels of correlations between the individual samples, and low detection rates of commonly mutated individual genes, should not be seen as evidence of failure, rather these are the expected result and should be seen as confirmation that a method such as INVAR is required for detection at such low levels, where individual mutations are not reliably detected.

To acknowledge the valid concerns raised by the Reviewer, we include the following sentence in the discussion to correct our interpretation in accordance with the reviewer's concern: "As a tumor-guided sequencing method, the accuracy of our approach is limited by the nature of the original samples and identified mutations with which we then used to design the capture panel. We have attempted to minimize contamination by mutations originating from clonal expansion in healthy tissues by collecting multiple tumor tissue subparts that were carefully selected during pathological examination. Subsequent tumor-guided sequencing studies will need to select normal tissue DNA as well as tumor tissue DNA in order to formally exclude the risk of cross-contamination from non-glioma mutations".

We would like however to highlight that:

1. We performed WES and capture sequencing on healthy controls during the technical validation of the INVAR method as previously published (Wan et al, 2020, STM and Smith et al, 2020, Genome Medicine). Also, the capacity of INVAR to filter out non-cancer variants at very low tumor fraction has been demonstrated previously on other cancer types, including early-stage cancer (cf Figure 3 and S4 from Wan et al, 2020, STM).
2. We compared the corresponding tumor tissue mutant allele fraction of patient specific variants against non-patient specific variants identified in fluids (see below figure). In all bio-fluids tested the patient specific variants corresponded to variants in tissue with a significantly higher mutant allele fraction (ranging from ~0.01 - ~0.25), in comparison to non-patient specific variants. Of note, if these variants corresponded to clonal expansions in 'other tissues' they would not be expected to have such high tumor allele fractions in the glioma tissue. In addition to this, we compared error-rates of the current data against those observed from a set of healthy plasma controls of a previous study (Wan et al., 2020, STM). This confirmed that the error-rate of these variants were not increased in control samples.

3. The observation that the size of mutant cfDNA in both plasma and urine of gliomas patients with INVAR is shorter than non-mutant cfDNA (Figure 4), as previously observed in other cancer types (Mouliere et al, 2018, STM; Wan et al, 2020, STM; Chabon et al, 2020). These results strongly suggest that the mutant ctDNA detected are matching the size expected from DNA released by cancer cells (and not normal tissue or white blood cells) in these biofluids.

We are therefore confident that a majority of CSF, plasma, and urine cfDNA fragments labelled as “tumor-derived” by INVAR (as described by using the parameters from the work in Wan et al, 2020), are indeed derived from cancer cells, thus supporting our general conclusions.

In addition, regarding the reviewer’s question on the age of the patients, the age is included in table EV1 – we have harmonized the sample naming on this table to clarify the link with the main figures in the text.

Second: the authors use shallow whole genome sequencing and analyses of overall DNA fragment length profiles to identify glioma based cfDNA. I had previously raised the issue that it is unlikely that the overall fragment length distribution is meaningfully affected by the presence of a tiny proportion of glioma specific DNA. Reviewer #3 raised a similar point "A second question arises about how the change in cfDNA fragmentation pattern, so remarkable in glioma patients vs. control individuals, can be reconciled with the extreme paucity of tumor DNA in blood and urine". The authors counter "Due to the non-targeted nature of sWGS, our approach recovers not only scarce cfDNA fragments with relevant SNVs, but also the majority of cfDNA fragments released by the tumor and those released by the adjacent brain". I remain unconvinced of this argument. The non-targeted assay recovers more tumor DNA of course, but it also recovers more DNA from healthy tissues. An alternative explanation here is that the shortening of cfDNA in the plasma of glioma patients compared to controls is due to some other change in patient physiology, it is unlikely the authors are detecting glioma specific DNA in this way.

We thank the Reviewer for this comment. In the manuscript, we are not claiming that our sWGS approach enables us to directly detect glioma-specific DNA in the urine samples and apologize if this was not clear in our previous reply. By using cfDNA fragmentation recovered from sWGS we can identify differences in the size profile of urine cfDNA from healthy, pathological and gliomas cases, and we can leverage this difference to classify glioma cases against a group of controls.

We agree that an extra word of caution in our interpretation from the sWGS data is necessary to clarify our claims. We have included the following paragraph in the discussion: “It is possible that the shortening of cfDNA in the urine of glioma patients compared to controls is due, at least in part, to differences in patient physiology and that this may directly contribute to the detection of a fragmentation-based glioma cfDNA signal in urine.” This sentence is added to another paragraph already present in our discussion: “Beyond the tissue of cancer origin, it is likely that urine cfDNA fragmentation might also be influenced by patient physiology (Teo et al, 2019), and pre-analytical parameters (Bosschieter et al, 2018). We attempted to mitigate for these effects by assessing the effect of age on the cfDNA fragmentation of urine samples, by controlling for the duration of pre-operative fasting, by using standardised sample preparation and DNA isolation and also by assessing the effect of tumor size on detectability.”

We have also corrected any potential sentence in the document that was unclear regarding our interpretation of the urine cfDNA fragmentation results.

We would like however to highlight that:

1. Reports highlight that 2-4% of cfDNA can originate from brain related cells in healthy individuals. This proportion can be even higher in certain pathological conditions. We would highlight that this is not a negligible fraction (Moss et al, 2019, Nature Communication), though concede that the majority of cfDNA in plasma and urine originate from hematopoietic and bladder epithelial cells respectively.
2. We agree that the impact of physiological (or clinical) variables could be a potential cause of the observed differences between the different samples. Indeed, we took steps to either rule out or estimate the role physiological variables could play in altering fragmentation patterns. We initially assessed the impact of age on the cfDNA fragmentation in urine samples (cf Figure S5) and observed no clear differences. We also investigated the effect of fasting which again highlighted no clear differences. We tried to minimize variations in the time of sampling and processing of the samples to exclude any potential technical confounding factors. We also included samples from other CNS pathologies to further increase the confidence that our observations are specific to cancer patients, and not only patients with brain pathologies. All these points were previously detailed in the discussion.

Referee #3 (Comments on Novelty/Model System for Author):

Concerning the medical impact, the authors provide only proof of concept that an innovative but complex and expensive methodology to detect glioma DNA in plasma and urine is feasible.

We thank the Reviewer for this comment. We agree that the tumor-guided sequencing approach employed here is a complex methodology, but as indicated by the reviewer, our objective was to demonstrate the proof of principle that urine could be used in glioma patients. We expect that further technological developments, combined with the decreasing cost of sequencing, could make this approach more accessible.

Referee #3 (Remarks for Author):

The authors adequately addressed the points raised by this reviewer and rewrote the manuscript in order to improve clarity and efficacy.

Please amend this inconsistency:

-line 838: AUC = 0.921 while in Fig. 6E AUC = 0.885.

We thank the Reviewer for this comment. This has been corrected in the revised document.

Please check nomenclature correspondence between column A in Table EV1 and in Table EV2 (Bling vs. GB)

We thank the Reviewer for this comment. The naming in EV1 has been corrected in the revised document.

7th Jun 2021

Thank you for the submission of your revised manuscript to EMBO Molecular Medicine. We have now received the enclosed report from referee #2. As you will see, this referee is supportive of publication, and I am therefore pleased to inform you that we will be able to accept your manuscript once the following editorial points will be addressed:

1/ Main manuscript text:

- Please answer/correct the changes suggested by our data editors in the main manuscript file attached (in track changes mode). Please use this file for any further modification.
- Please remove the red text.
- Please remove the references to Source Data in the main manuscript text.
- Material and methods:
 - o Study design: please include the full statement that informed consent was obtained from all subjects and that the experiments conformed to the principles set out in the WMA Declaration of Helsinki and the Department of Health and Human Services Belmont Report.
 - o Please add a "Statistical analyses" section.
- Thank you for providing a Data Availability section. Please provide a direct link to access the data and note that the data have to be publicly available before acceptance of the manuscript.

2/ An ORCID identifier is still missing for Richard Mair. We can unfortunately not link his profile to an ORCID identifier ourselves, it has to be done by the author himself.

3/ As part of the EMBO Publications transparent editorial process initiative (see our Editorial at <http://embomolmed.embopress.org/content/2/9/329>), EMBO Molecular Medicine will publish online a Review Process File (RPF) to accompany accepted manuscripts.

This file will be published in conjunction with your paper and will include the anonymous referee reports, your point-by-point response and all pertinent correspondence relating to the manuscript. Let us know whether you agree with the publication of the RPF and as here, if you want to remove or not any figures from it prior to publication.

I look forward to receiving your revised manuscript.

With my best wishes,

Lise

Lise Roth, PhD
Editor
EMBO Molecular Medicine

To submit your manuscript, please follow this link:

Link Not Available

The system will prompt you to fill in your funding and payment information. This will allow Wiley to send you a quote for the article processing charge (APC) in case of acceptance. This quote takes into account any reduction or fee waivers that you may be eligible for. Authors do not need to pay any fees before their manuscript is accepted and transferred to our publisher.

***** Reviewer's comments *****

Referee #2 (Remarks for Author):

The authors have adequately addressed my remaining concerns, and I am now happy to recommend this manuscript for publication in EMBO Molecular Medicine.

The authors performed the requested editorial changes.

14th Jun 2021

Thank you for submitting your finalized manuscript.

I am pleased to inform you that your article is now accepted for publication in EMBO Molecular Medicine!

I have accepted all the changes, and I completed the sentence "the experiments conformed to the principles set out in the WMA Declaration of Helsinki and the Department of Health and Human Services Belmont Report". Please let us know immediately if you do not agree with it.

Please also be reminded that all data have to be made public before publication of your manuscript.

Follow us on Twitter @EmboMolMed

Sign up for eTOCs at embopress.org/alertsfeeds

*** ** IMPORTANT INFORMATION ** **

SPEED OF PUBLICATION

The journal aims for rapid publication of papers, using using the advance online publication "Early View" to expedite the process: A properly copy-edited and formatted version will be published as "Early View" after the proofs have been corrected. Please help the Editors and publisher avoid delays by providing e-mail address(es), telephone and fax numbers at which author(s) can be contacted.

Should you be planning a Press Release on your article, please get in contact with embomolmed@wiley.com as early as possible, in order to coordinate publication and release dates.

LICENSE AND PAYMENT :

All articles published in EMBO Molecular Medicine are fully open access: immediately and freely available to read, download and share.

EMBO Molecular Medicine charges an article processing charge (APC) to cover the publication

costs. You, as the corresponding author for this manuscript, should have already received a quote with the article processing fee separately. Please let us know in case this quote has not been received.

Once your article is at Wiley for editorial production you will receive an email from Wiley's Author Services system, which will ask you to log in and will present you with the publication license form for completion. Within the same system the publication fee can be paid by credit card, an invoice, pro forma invoice or purchase order can be requested.

Payment of the publication charge and the signed Open Access Agreement form must be received before the article can be published online.

PROOFS

You will receive the proofs by e-mail approximately 2 weeks after all relevant files have been sent to our Production Office. Please return them within 48 hours and if there should be any problems, please contact the production office at embopressproduction@wiley.com.

Please inform us if there is likely to be any difficulty in reaching you at the above address at that time. Failure to meet our deadlines may result in a delay of publication.

All further communications concerning your paper proofs should quote reference number EMM-2020-12881-V4 and be directed to the production office at embopressproduction@wiley.com.

Thank you,

Lise Roth, Ph.D
Scientific Editor
EMBO Molecular Medicine

Corresponding Author Name: F.Mouliere
Journal Submitted to: EMBO Molecular Medicine
Manuscript Number: EMM-2020-12881